Corrected: Author Correction

# SREBP1 drives Keratin-80-dependent cytoskeletal changes and invasive behavior in endocrine-resistant ERα breast cancer

Ylenia Perone [1,14], Aaron J. Farrugia[2,14], Alba Rodríguez-Meira[1,12,14], Balázs Győrffy[3,4], Charlotte Ion[1], Andrea Uggetti[5], Antonios Chronopoulos[6], Pasquale Marrazzo [7], Monica Faronato[8], Sami Shousha[9], Claire Davies[10], Jennifer H. Steel [10], Naina Patel[10], Armando del Rio Hernandez [6], Charles Coombes[1], Giancarlo Pruneri[11], Adrian Lim[1], Fernando Calvo[2,13] & Luca Magnani [1]

Approximately 30% of ERα breast cancer patients relapse with metastatic disease following adjuvant endocrine therapies. The connection between acquisition of drug resistance and invasive potential is poorly understood. In this study, we demonstrate that the type II keratin topological associating domain undergoes epigenetic reprogramming in aromatase inhibitors (AI)-resistant cells, leading to Keratin-80 (KRT80) upregulation. KRT80 expression is driven by de novo enhancer activation by sterol regulatory element-binding protein 1 (SREBP1). KRT80 upregulation directly promotes cytoskeletal rearrangements at the leading edge, increased focal adhesion and cellular stiffening, collectively promoting cancer cell invasion. Shearwave elasticity imaging performed on prospectively recruited patients confirms KRT80 levels correlate with stiffer tumors. Immunohistochemistry showed increased KRT80-positive cells at relapse and, using several clinical endpoints, KRT80 expression associates with poor survival. Collectively, our data uncover an unpredicted and potentially targetable direct link between epigenetic and cytoskeletal reprogramming promoting cell invasion in response to chronic AI treatment.

[1] Department of Surgery and Cancer, Imperial College London, London, UK. [2] Division of Cancer Biology, Tumour Microenvironment Team, Institute of Cancer Research, London, UK. [3] MTA TTK Lendület Cancer Biomarker Research Group, Institute of Enzymology, Hungarian Academy of Sciences, Budapest, Hungary. [4] 2nd Department of Pediatrics, Semmelweis University, Budapest, Hungary. [5] European Institute of Oncology, Milan, Italy. [6] Faculty of Engineering, Department of Bioengineering, Imperial College London, London, UK. [7] Department for Life Quality Studies, Alma Mater Studiorum, University of Bologna, Rimini, Italy. [8] Department of Chemistry, Imperial College London, London, UK. [9] Histopathology Department, Imperial College London, Charing Cross Hospital NHS Trust, London, UK. [10] ECMC Imperial College. Department of Surgery and Cancer, Imperial College London, London, UK. [11] Pathology Department, Fondazione IRCCS Istituto Nazionale Tumori and University of Milan, School of Medicine, Milan, Italy. [12] Present address: MRC Molecular Haematology Unit, Haematopoietic Stem Cell Biology Laboratory, Weatherall Institute of Molecular Medicine, University of Oxford, Oxford, UK. [13] Present address: Instituto de Biomedicina y Biotecnologia de Cantabria, Santander, Spain. [14] These authors contributed equally: Ylenia Perone, Aaron J. Farrugia, Alba Rodríguez Meira. Correspondence and requests for materials should be addressed to F.C. (email: calvof@unican.es) or to L.M. (email: l.magnani@imperial.ac.uk)

Aromatase inhibitors (AI) treatment is standard of care for breast cancer (BC), yet BC cells frequently display drug-resistance and stronger metastatic potential at relapse, suggesting that chronic exposure to endocrine treatment might contribute in shaping the invasive potential, as suggested by previous in vitro studies[1,2]. The mechanism/s, order of events and molecular players mediating these phenomena are not well understood but it is likely that they involve cytoskeletal re-arrangements as they are essential for cancer invasion and metastasis[3]. One possibility is that endocrine therapies (ET) might indirectly promote invasive behaviors by selecting for interrelated phenotypes during tumor evolution[4–6]. Alternatively, AI treatment may directly contribute to the activation of invasive transcriptional programs. Chronic exposure to ET leads to coordinated activation and decommissioning of regulatory regions such as enhancer and promoters as shown by global changes in the localization of epigenetic marks H3K27ac and H3K4me1-2[6,7,8]. These epigenetic changes occasionally involve entire topological associating domains (TADs), three-dimensional compartments within the genome thought to restrict enhancer-promoter interactions[9,10]. In this manuscript, we show how drug-induced epigenetic reprogramming leads to significant cytoskeletal changes and mechano-properties at the cellular level to promote invasive behavior.

## Results

**Epigenetic reprogramming leads to KRT80 expression in drug-resistant BC.** We have previously shown that the type II keratin TAD[7] ranked among the most significantly epigenetically reprogrammed TADs when comparing untreated (MCF7, ERα-positive breast cancer cell lines) non-invasive ET-treated (MCF7 cells resistant to Tamoxifen: MCF7T or Fulvestrant: MCF7F) vs. invasive AI-resistant BC cell lines[7] (MCF7 that were long term estrogen deprived: LTED cells, and double resistant LTEDT and LTEDF Fig. 1a, b). ChIP-seq efficiencies were rather different across each cell line but, genome-wide normalization confirmed that overall, the type II keratin TAD accrues significantly more H3K27ac reads in invasive LTED cells compared with MCF7 and MCF7T cells (Top 5% for differential[9], Fig. 1a inset). Targeted validation within one of the potential enhancers (E1) using H3K27ac, H3K4me2, and H3K4me1 confirmed the significant increase of H3K27ac between MCF7 and LTED (Fig. 1c). Type I and Type II Keratins are the main constituents of cytoplasmic intermediate filaments and are involved in crucial cellular processes including cell attachment, stress adaptation, and cell structure maintenance; yet very little is known about their role in cell movement and metastatic progression. Despite TAD dynamics, only few keratins within the type II-keratin TAD were transcriptionally reprogrammed in AI-resistant cell lines, with KRT80 being the only member which was consistently upregulated in all LTED models, including LTED-derivatives from a different breast cancer cell line (T47D, Fig. 1a, b and Supplementary Fig. 1a, b). Live-tracking cells during the initial 48 h of estrogen deprivation shows the absence of substantial proliferation and/or cell death, suggesting that the majority of cells simply stall within this time frame (Fig. 1d, flat orange line from 6 to 48 h). Measuring KRT80 transcripts before or after short-term (48 h) acute estrogen starvation using single cell RNA-seq data shows a significant increase in the proportion of KRT80 positive cells, strongly suggesting that this increase is driven by de novo transcriptional activation and not selection of KRT80-positive clones (Fig. 1e). These data were validated in MCF7 and LTED cells using single cell RNA-FISH (Fig. 1f). As expected, increased transcription corresponded to increased KRT80 protein level in both MCF7 and T47D models (Fig. 1g). Interestingly, LTED

cells also show significant changes in H3K27ac levels and mRNA expression for cholesterol biosynthesis genes[7], but unexpectedly the master regulator of cholesterol biosynthesis SREBP1[11] shows no transcriptional changes between the two cell types, suggesting that reprogramming is not driven by transcriptional factor abundance but rather by its activity[7] (Fig. 1f).

**KRT80 dynamically changes during breast cancer progression in vivo.** KRT80 is a largely unknown keratin structurally related to hair keratins[12], in contrast with epithelial keratins commonly found in normal epithelial cells. This led us to further explore the role of KRT80 in promoting the invasive phenotype developed by LTED as a consequence of AI-resistance[7]. KRT80 transcripts were also elevated in several ERα-negative cell lines, suggesting that upregulation in drug-resistant cells was not mediated by changes in ERα activity (Supplementary Data 1). More importantly, IHC analysis of two independent clinical datasets confirmed that KRT80 positive cells significantly increase after AI treatment while showing a trend in Tamoxifen-treated patients in vivo[13,14] (Fig. 2a). KRT80 localization in vivo was radically different to what has been shown in conventional keratins (e.g., KRT8, KRT14, KRT18, or KRT19[15]), presenting a peri-nuclear polarized pattern towards the lumen within healthy ducts and lobules (Fig. 2b). Similar staining patterns were conserved in benign lesions (Fig. 2b), whereas KRT80 staining became strongly cytoplasmic in higher grade BC and metastatic lesions suggesting a potential role in BC progression (Fig. 2b). Correspondingly, high KRT80 mRNA levels correlated with poor survival in the METABRIC ERα-positive BC dataset (Fig. 2c), even more significantly when selecting patients that did relapse early and were treated with endocrine therapies (Fig. 2c). The prognostic role of KRT80 was then confirmed by multivariate meta-analysis of two independent datasets with several additional clinical endpoints (Supplementary Fig. 1c–e and Supplementary Fig. 2). Interestingly, KRT80 was the only reprogrammed Type-II keratin significantly associated with clinical endpoints in BC patients (Fig. 2d).

**De novo SREBP1 drives KRT80 activation.** Activation of cell type specific enhancers has been linked with cancer transcriptional aberration[6,16–18], leading us to hypothesize that de novo enhancer activation within the TAD structure might control KRT80 expression in AI resistant cells. We used H3K27ac, an epigenetic mark associated with gene activation[6,19], to narrow down the potential KRT80 enhancers (E1 and E2, Fig. 1b). As expected, E1-E2 activity was only captured in KRT80-positive cells (Fig. 1b) while E1 enhancer activity analysis predicted a significant increase in KRT80 positive cells in AI resistant models (Fig. 1c), in agreement with mRNA and protein analysis (Fig. 1a, f, g and Supplementary Fig. 3a). 3D meta-analysis from parental MCF7 ChIA-Pet data strongly suggested that the E1 loci could contact the KRT80 promoter via enhancer-promoter interactions, while it excluded the weaker E2 (Supplementary Fig. 3b) suggesting that the 3D interaction is already pre-established in sensitive cells. To test whether E1 drove KRT80 transcriptional activity in other context, we adapted our recently developed computational pipeline to measure the relative size of KRT80-positive clones in several tissues[6]. This pipeline can estimate the percentage of cells containing an active enhancer, as at individual loci the epigenetic signal is a function of the number of modified nucleosomes[6]. We thus tested if the estimated size of KRT80 positive cells based on E1 activity in each model is reflected at the transcriptional level. Analysis of Epigenetic Roadmap data with associated transcriptional profiles[20] strongly suggested that increasing E1 positivity, predicting for increasing content of

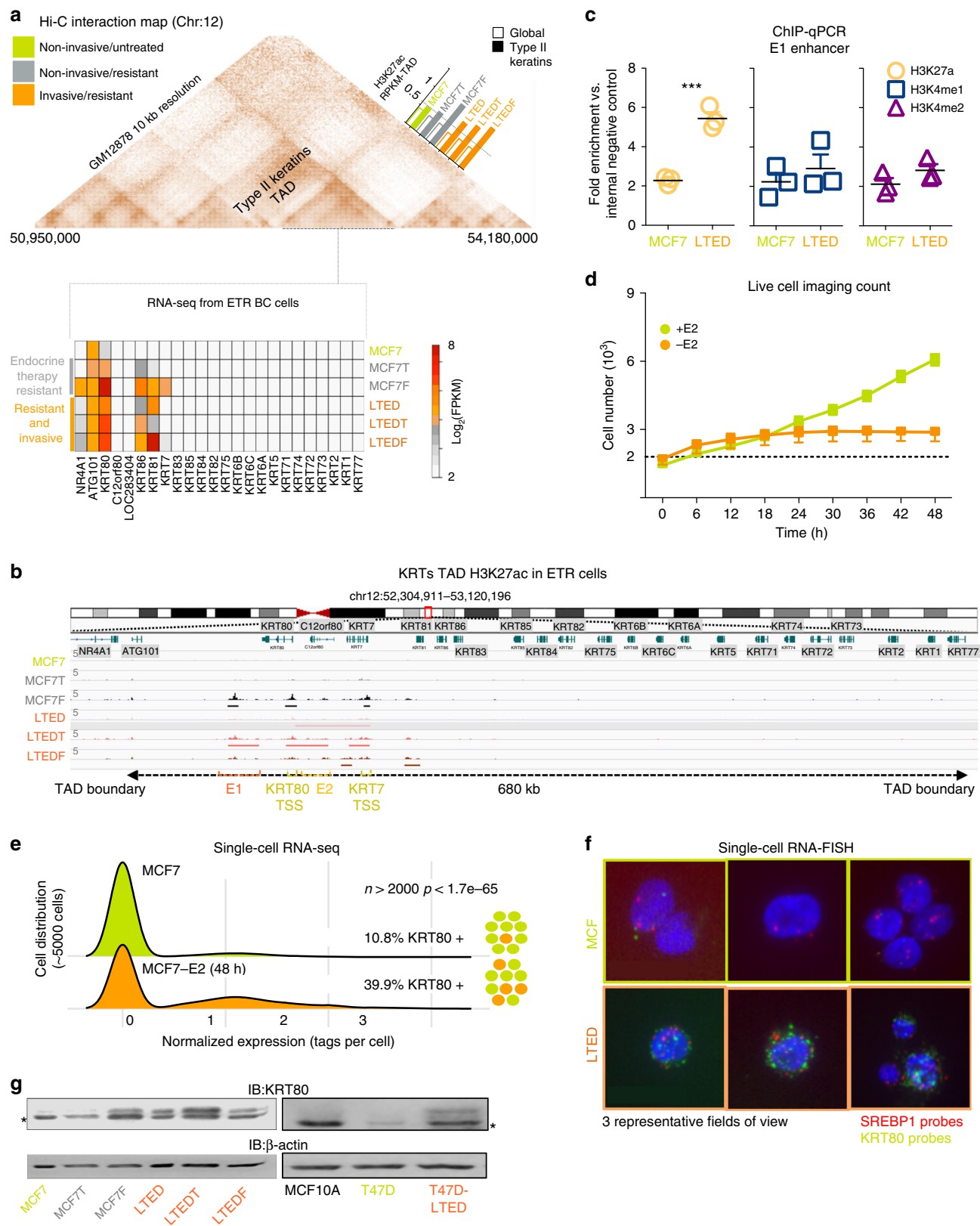

KRT80-positive cells, correlates KRT80 transcription levels (Fig. 3a). E1 activity was also potentially associated with KRT80 transcription in several cell lines (Supplementary Fig. 3c, d). For example, Keratinocytes ranked as the most clonal KRT80 cell type and exhibited the highest KRT80 mRNA levels (Supplementary Fig. 3c, d). Colon cancer HCT116 cells also were predicted to contain a clonal KRT80 cell population based on E1 activity (Supplementary Fig. 3c, d). On the other hand, E1 predicts only for a small subpopulation within normal cells from the large intestine (Fig. 3a). Interestingly, KRT80 is dramatically upregulated during intestine oncogenesis[21] (ranked 4th overall as the most significantly upregulated gene in TCGA normal vs. cecum

**Fig. 1** AI treatment induces KRT80 expression via epigenetic reprogramming. **a** Hi-C 3D interactions in GM12878 cells were analyzed using http://promoter.bx.psu.edu/hi-c/view.php. Data to derive individual TAD were downloaded from http://chromosome.sdsc.edu/mouse/hi-c/download.html. Bars represent the normalized median change in H3K27ac within the Type II-Keratin TAD compared to the overall change in H3K27ac between parental MCF7 cells (green) and drug-resistant non-invasive (gray) and drug-resistant invasive (orange) counterparts. The bottom heatmap shows the normalized expression of RNA-seq data for protein coding genes within the Type II-Keratin in all breast cancer cell lines. **b** Bird-eye view of the H3K27ac profile of the Type II-Keratins locus. ChIP-seq signal profiles from[7] are shown across the entire TAD. **c** Targeted ChIP-qPCR for the E1 enhancer locus using H3K4me1, H3K4me2 and H3K27ac antibodies. Individual biological replicates, mean and SD are shown. Asterisks represent significance at the $p < 0.001$ level. **d** Live-imaging cell counts of mate-labeled MCF7 cells grown in presence or absence of estrogen for 48 h. Dotted line represents an ideal stalling dynamic in cell number during the time of the assay. Mean and SD of three independent counts are shown. **e** Population level single-cell RNA-seq data for KRT80 expression are shown. KRT80 was identified in 10.8% of MCF7 cultured in estrogen rich media and in 39.9% of MCF7 deprived of estrogen for 48 h. The distribution of the two set of data was compared using a Fisher exact test. Experiments were run comparing cells within 48 h in absence of major cell division/apoptosis. **f** Representative single-molecule, single cell RNA-FISH for SREBP1 (red) and KRT80 (green) in MCF7 and LTED cells. **g** KRT80 protein levels in MCF7 and additional independent models of invasive drug-resistant breast cancer cell lines. The asterisk represents an unspecific band. Acronyms: LTED cells: MCF7 that were long term estrogen deprived; and double resistant; MCF7T, MCF7F, LTEDT, and LTEDF: MCF7 and LTED cells resistant to Tamoxifen or to Fulvestrant respectively; ETR: endocrine treatment resistant; E2: estrogen; TAD: topological associated domain; RPKM: reads per kilobase million; KRT: keratin; SREBP1: sterol regulatory element binding protein 1

cancer) strongly suggesting a progressive activation of E1 during colon cancer transformation. E1 also predicts for clonal KRT80 expression in HUVEC cells and HUVEC cells are characterized by strong KRT80 transcription (Supplementary Fig. 3c, d). Finally, KRT80 E1 activity also correctly predicted strong expression in mammary epithelium cells (Supplementary Fig. 3c, d). Conversely, samples with no E1 activity were found to have no KRT80 transcription (i.e., immune cells and iPS cells). Overall these data strongly link E1 to KRT80 transcription. As E1 enhancers span nearly 12.5 Kb, we performed fine-mapping analysis to narrow down on potential readers. Using our computational pipeline, we sought for E1 sub-regions more strongly associated with KRT80 expression in our BC cell lines leading to the identification of a core-region within the E1 enhancer (1.5 Kb) (Supplementary Fig. 3a). This core enhancer showed a clear pattern of activity in actual BC patients[6] predicting the existence of KRT80 clonal and sub-clonal populations in primary and metastatic BC (Fig. 3b). We next investigated which transcription factor/s (TFs) might regulate KRT80 expression via core-E1 binding. DHS-seq analysis[7] indicated that KRT80 is already accessible in MCF7 (Fig. 3c), yet digital foot-printing suggested different occupancy sites (Fig. 3d). Intriguingly, among other footprints, we noted the appearance of a SREBP1 footprint within the core-E1 unique to LTED cells. We have previously reported that AI resistant cells upregulate lipid biosynthesis via global epigenetic reprogramming[7] suggesting widespread SREBP1 activation in AI resistant cells. However, SREBP1 is not differentially expressed in LTED cells when compared with parental MCF7 cells (Fig. 1f), suggesting that SREBP1 might upregulate its targets by increased nuclear shuttling and chromatin binding. This led to the hypothesis that increased SREBP1 occupancy might drive KRT80 transcriptional activation in LTED cells. ENCODE TFs mapping showed that SREBP1 can bind the core-E1 enhancers in lung cancer cells, the only ENCODE profiled cells characterized by strong KRT80 transcription (Supplementary Fig. 4a, b). To directly test if SREBP1 drives KRT80 expression in BC we performed ChIP-seq in MCF7 and T47D cells and their respective AI-resistant models. Our data demonstrate that SREBP1 was bound at core-E1 only in AI-resistant BC cells (Fig. 3e and Supplementary Fig. 4c). Interestingly, the expression of KRT80 and SREBP1 target genes was also strongly correlated in BC patients (Supplementary Fig. 4d). Finally, we show that SREBP1 silencing abrogated KRT80 expression in LTED cells (Fig. 3f, g). Overall these data demonstrate an unpredicted link between SREBP1 and KRT80 activation. Phastcons, PhyloP and Siphy rates, which measure the rate of DNA conservation between different species, show a significant drop in conservation

at the SREBP1 footprint within the otherwise conserved E1 enhancer (Supplementary Fig. 5), suggesting that the link between SREBP1 and KRT80 might have evolved relatively recently. Overall, these data strongly support the hypothesis that the core-E1 is the critical enhancer driving KRT80 expression in BC cells.

**KRT80 directly promotes increased tumor stiffness in vitro and in vivo.** Several studies have investigated how mechanical stimuli influence the epigenetic landscape[22,23]. However, our data implied a novel causal link whereby epigenetic reprogramming promoted changes in specific cytoskeletal components (e.g., KRT80) which may ultimately affect the biophysical properties of cells and tumors[24,25] (Figs. 1–3). In agreement, we observed a significant increase in cellular stiffness (inversely correlated to cell compliance/deformability) at the single cell level after KRT80 over-expression in MCF7 and LTED cells (Fig. 4a). Conversely, KRT80 depletion in LTED cells resulted in a significant loss of cellular stiffness (Fig. 4a). To test if KRT80 can contribute to tumor stiffness in vivo we prospectively recruited 20 patients with suspected BC and performed shear-wave elastography to measure intra-tumoral stiffness. Elastography was performed prior to biopsies were taken but all cases were subsequently confirmed positive breast cancer (Fig. 4b). Our data showed that cancer lesions had significantly higher stiffness than surrounding normal tissues, with the highest peak of stiffness consistently measured at the invasive border (Fig. 4b). Interestingly, meta-analysis of tumor and matched nearby tissue from TCGA show increased KRT80 mRNA in the tumor biopsies (Supplementary Fig. 6). We then performed IHC for KRT80 with validated antibodies (Fig. 4c and Supplementary Fig. 7) using biopsies collected from our prospective patients. Linear regression analysis showed that KRT80 positivity significantly correlated with intra-tumor stiffness (Fig. 4d). Collectively, these data demonstrate that BCs characterized with high KRT80 content are mechanically stiffer.

**KRT80 upregulation leads to augmented collective invasion.** The effect of increasing stiffness in metastatic invasion is highly debated. Previous studies have suggested that decreased stiffness, through loss of keratins, improves single-cell invasion[24] typical of EMT cells. However, solid tumors can also use a myriad of multicellular invasion programs[26] collectively termed "collective invasion". Recent studies have shown that keratins such as KRT14 can play critical roles in collective invasion[27] and multiclonal metastatic seeding[27,28], two processes driving BC progression[27]. In addition, a significant body of clinical literature has linked increased breast tumor stiffness to poorer prognosis[27,29–31]

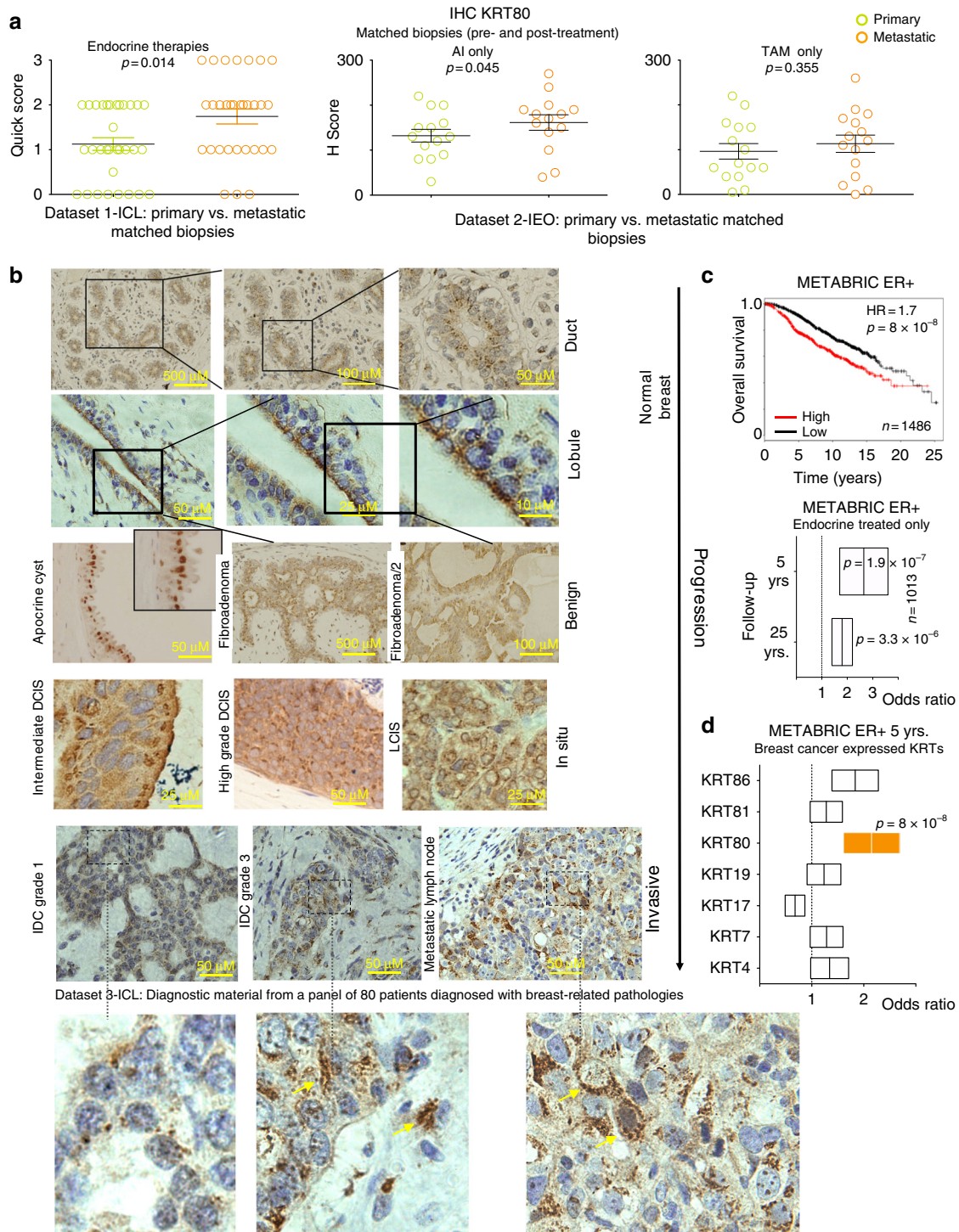

**Fig. 2** KRT80 dynamics in treated progressing breast cancer patients. **a** Matched clinical specimens from breast cancer patients show an increase in KRT80 positive cells following mono-treatment (Dataset 1-ICL: Imperial College London, UK) or sequential treatment with aromatase inhibitors (Dataset 2-IEO: Istituto Europeo di Oncologia, Milan, Italy). Similar results were not significant in Tamoxifen-only treated patients. **b** Immunocytochemistry (IHC) analyses show changes in KRT80 protein distribution. KRT80 was imaged using IHC in a series of human samples collected at Charing Cross Hospital Imperial College NHS Trust (ICL London, UK). Tissues were collected to cover a large spectrum of benign and malignant lesions including metastatic samples from Breast Cancer patients. Yellow arrows in the bottom panels highlight cells with KRT80 expanded cytoplasmic staining. **c** KRT80 expression in diagnostic material has prognostic significance. Analysis were performed on the METABRIC RNA-seq splitting patient in high and low KRT80 expression. Two distinct follow-ups for an additional sub-cohort is also shown (endocrine-treated patients). Floating bars show minimum-maximum and average hazard ratios. **d** Other transcribed type-II Keratins in breast cancer samples from METABRIC are not associated with prognostic significance. Floating bars show minimum-maximum and average hazard ratios

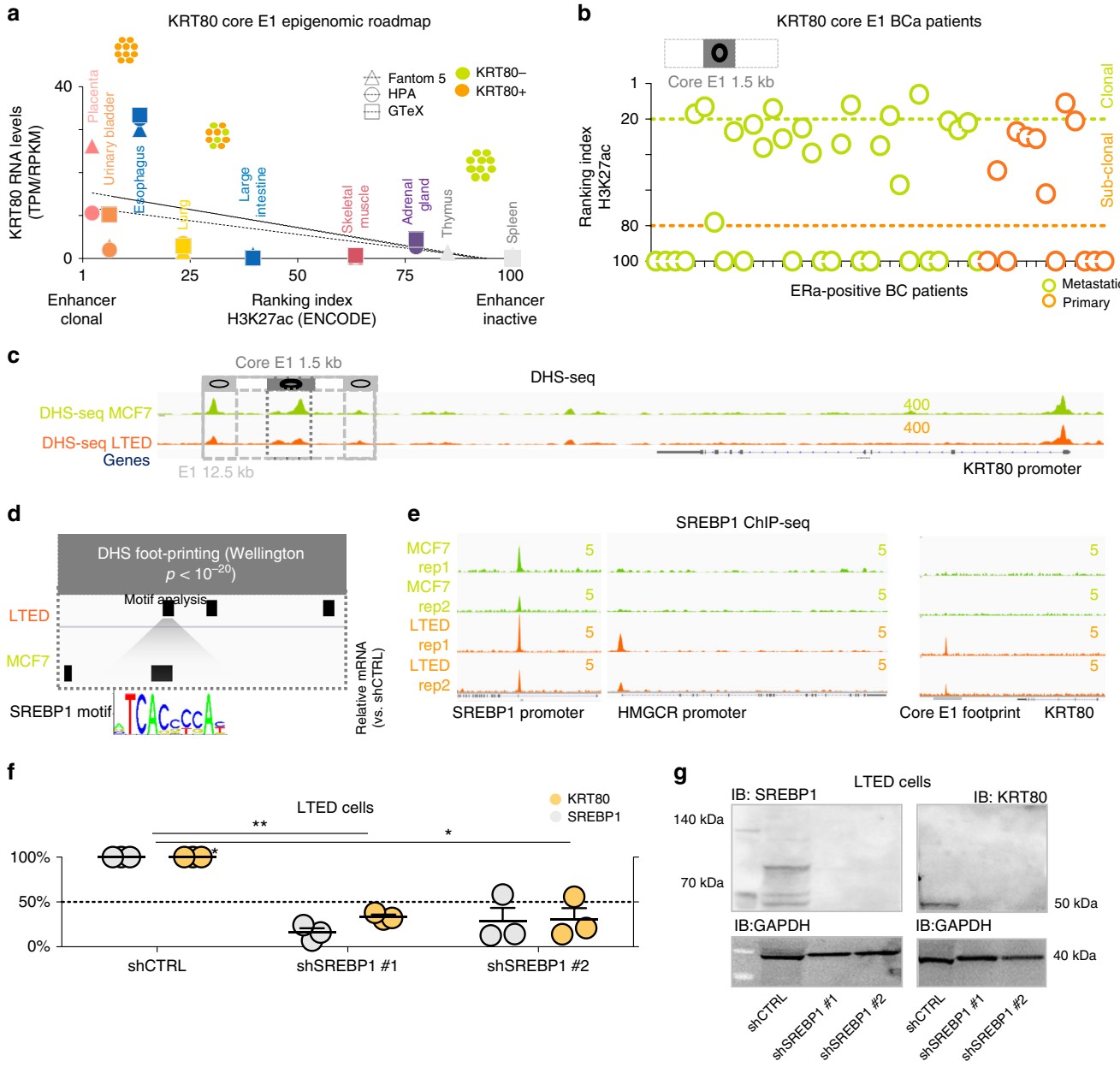

**Fig. 3** De novo SREBP1 binding at KRT80 enhancer drives KRT80. **a** Predicted KRT80 enhancer clonality (*x*-axis) and KRT80 RNA levels (*y*-axis) are plotted for three independent transcriptional datasets (fantom5; HPA, Human Protein Atlas, GTEx, Genotype-Tissue Expression). Epigenetic data were obtained from the ENCODE consortia. Increasing KRT80 enhancer clonality is associated with an increasing number of KRT80 positive cells (symbols at the top, see methods for more details). **b** Enhancer clonality from H3K27ac data obtained in primary and metastatic breast cancer biopsies. Green dotted line indicates the presence of clonal KRT80-positive lesions. Orange dotted lines predict for KRT80-low lesions. Each circle represents an individual patient. KRT80 clonality was calculated using the core 1.5Kb H3K27ac peak. **c** Open chromatin profiling via DHS-seq in MCF7 and LTED cells near the KRT80 locus. **d** Digital Foot-printing analysis shows differential occupancy status within the E1 KRT80 enhancer. Footprint were identified using Wellington with a $p < 10^{-20}$ threshold. **e** ChIP-seq analysis for SREBP1 at the E1 core enhancers in invasive AI resistant breast cancer cells and treatment naive parental cell lines. SREBP1 canonical target HMGCR locus is also shown. The SREBP1 locus is bound in both MCF7 and LTED and represent the only genomic location with SREBP1 binding in parental MCF7 cells. **f** Stable shSREBP1 silencing in LTED cells using two independent shRNA. Individual biological replicates are shown. Lines represent means and SD. Asterisk represent significant difference at $p < 0.05$ after One Way ANOVA with Dunnet's test. **g** Stable shSREBP1 LTED cells were assessed for SREBP1 and KRT80 protein levels

and lymph node positivity[27,29,30], independently of changes in extracellular matrix stiffness. We reasoned that a model in which KRT80 upregulation in BC cells leads to increased stiffness and augmented collective invasion might reconcile all these observations. To test this, we developed 3D spheroids from MCF7 or LTED cells and assessed collective invasion (Fig. 5a) after KRT80 manipulation (Supplementary Fig. 8a–d). Spheroids from

KRT80-positive LTED cells could effectively invade intro matrigel matrices whereas stable or transient KRT80 depletion completely abrogated the invasive phenotype (Fig. 5c, d and Supplementary Fig. 8e, f). Conversely, ectopic expression of KRT80 conferred matrix invading capacities in otherwise non-invasive MCF7 cells, even in the absence of chronic estrogen deprivation (Fig. 5b, d). KRT80 immunostaining showed that KRT80 positive cells

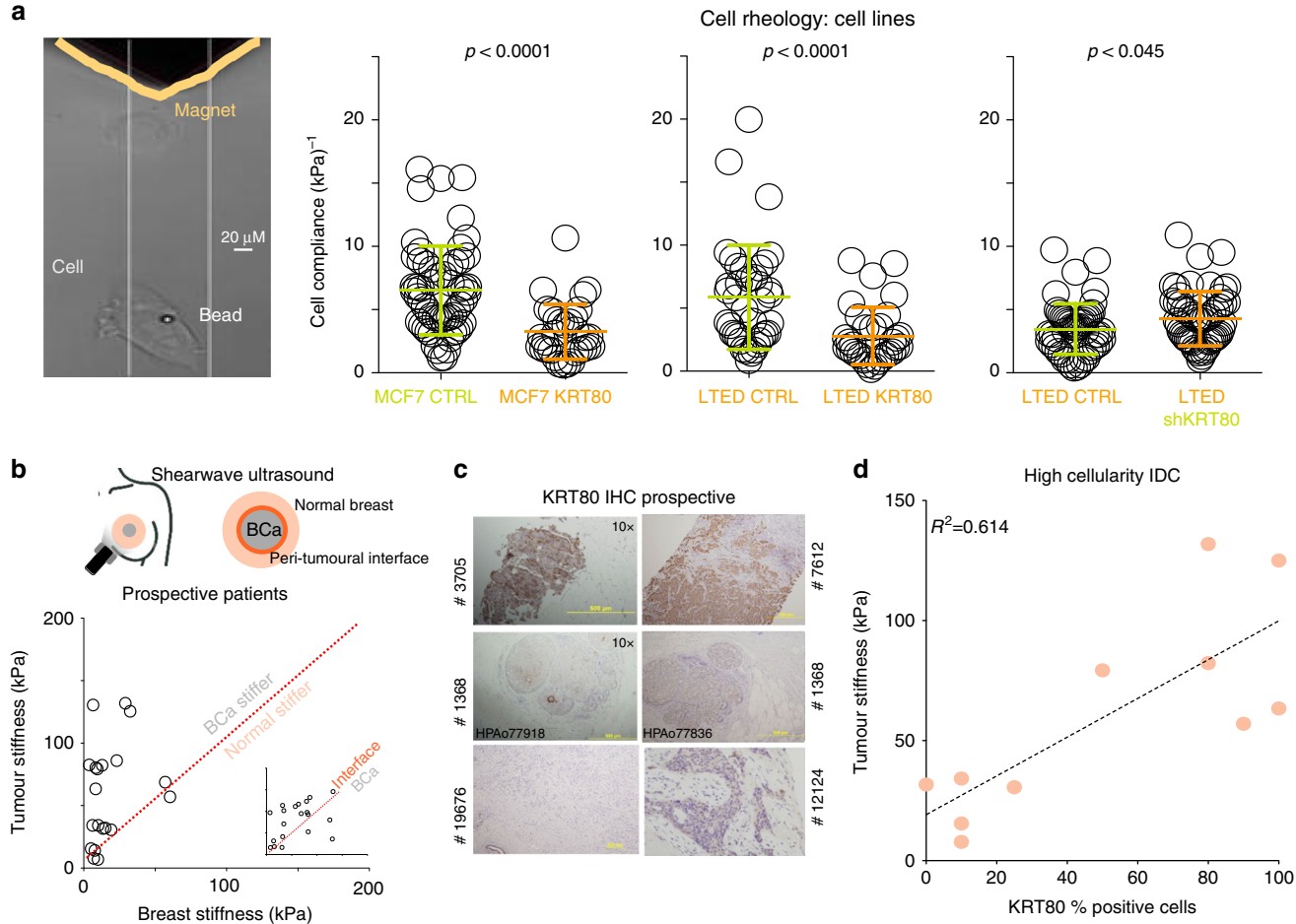

**Fig. 4** KRT80 levels are associated with changes in cell stiffness. **a** Magnetic tweezers (yellow) were used to measure the biomechanical properties of individual cells with or without KRT80 manipulation. Changes in cell compliance (deformation) were measured in KRT80 over-expressing cells for both parental MCF7 and AI-resistant LTED cells (left and middle graphs). Changes in cell compliance (deformation) were measured after stable KRT80 depletion in LTED cells (right graph). Significance was calculated with a student t- test and reported. **b** Shearwave Ultrasound measurements in prospectively recruited patients. Measures were collected at three independent location for each patient (see diagram). Plots show matched tissue stiffness for cancer vs. normal (large panel) and peri-tumoral interface vs. cancer (small inset). **c** KRT80 cells in diagnostic material from prospective patients assessed with ultrasound were counted using IHC. **d** Plots show matched tissue stiffness against the percentage of KRT80-positive cells for each individual patient. Simple linear regression was applied to calculate the correlation coefficient between these two values

clustered at the invasive front in LTED spheroids (Fig. 5e and Supplementary Fig. 9a, b), a pattern reminiscent of the leading cells characterized in epithelial tumors during collective invasion[27,28]. To confirm that invasion was driven by active motion rather than proliferation at the border of the organoids, we repeated invasion assays using proliferation sensitive live-labeling (Fig. 5f). Labeled cells maintained their invasive properties while KRT80 suppression still blocked invasion (Fig. 5g). As expected, invading cells retained the dye suggesting that they actively moved into the matrigel interface in absence of cell division (Fig. 5h). These data are supported by live-imaging of organoid invasion performed previously in the same cell lines[7].

**KRT80 reorganizes cells cytoskeleton to promote lamellipodia formation.** Confocal microscopy analyses informed that LTED and MCF7-KRT80 cells presented an intricate network of KRT80 filaments that significantly overlap actin fibers (Fig. 6a, b). This KRT80 network was prominent at the leading edge of cells, usually localized at or annexed to actin-rich lamellipodium-like structures (Fig. 6b, asterisk). Conversely, in KRT80low cells (i.e., MCF7 and LTED-shA), KRT80 staining was more punctuated

and mainly observed towards the cell cortex, with border cells presenting strong cortical actin (Fig. 6b, hashtag) and no prominent lamellipodia[32]. Quantitative analysis of confocal data showed that KRT80 expression was associated with a significant increase of F-actin at lamellipodial structures, with smaller compensating changes at the cell cortex and cytosol depending on the system (i.e., MCF or LTED) (Fig. 6c, d). Importantly, no significant changes were observed in the total F-actin between MCF7/MCF-KRT80 or LTED/LTED-shKRT80 (Fig. 6d). Together, these results suggest that the generation of a network of KRT80 positive filaments do not affect actin polymerization but rather reorganize the actin cytoskeleton to promote lamellipodia formation. In agreement, cells expressing KRT80 presented a higher proportion of cells with lamellipodia when compared with their KRT80low counterparts (Fig. 6e). Focal adhesion growth and maturation are tightly coupled with the forward movement of the lamellipodium[33], are associated to cell stiffness/cellular tension[29,30], and are particularly relevant in the generation of forces required for migration and invasion in complex settings. In line with KRT80 playing a role in these processes, we observed that KRT80 directly promoted the generation of larger more mature paxillin focal adhesions, with no significant change in the

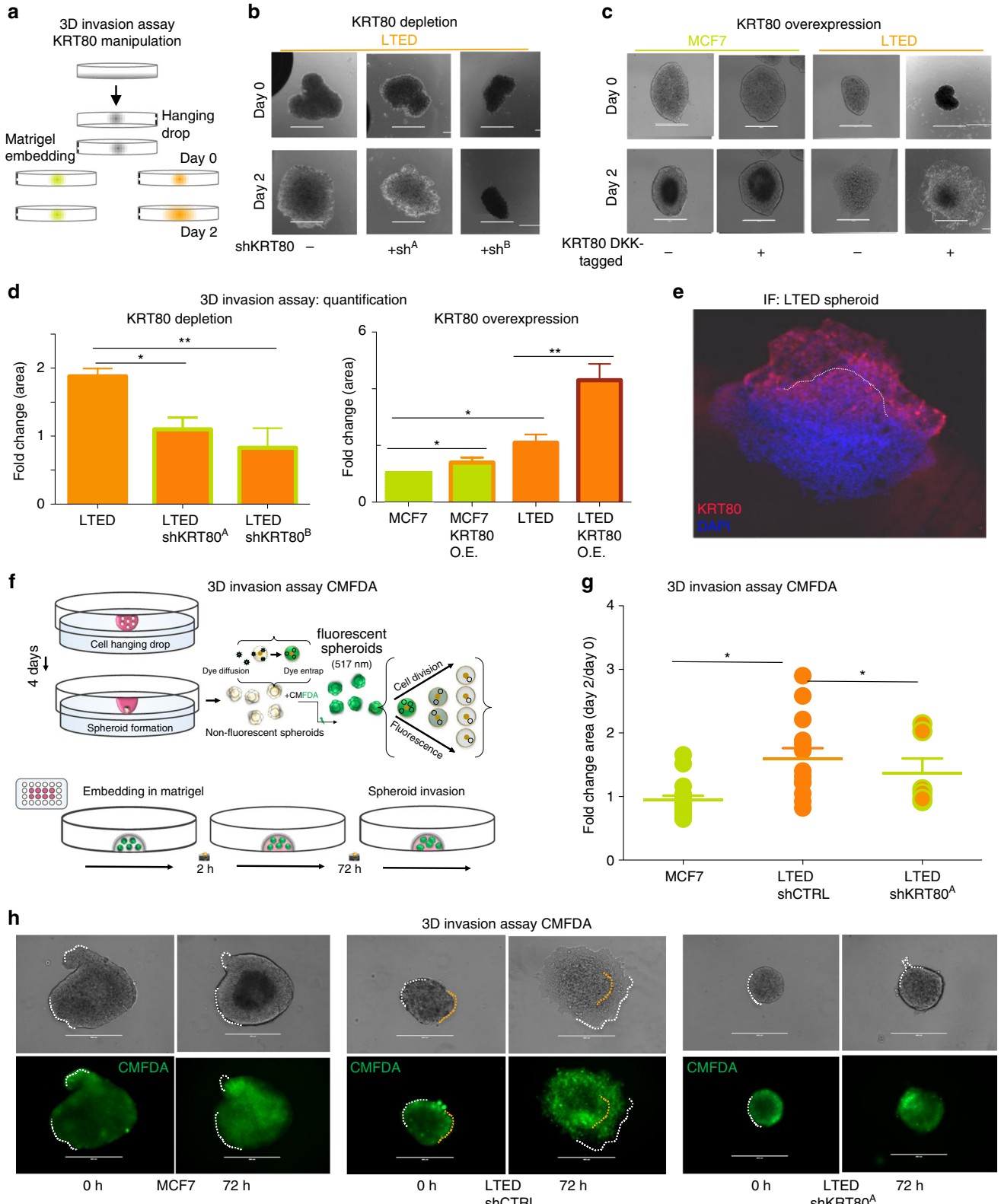

number of focal adhesions per cell (Fig. 6f). Interestingly, KRT80 positivity strongly characterized invading cells from prospectively collected pleural effusion from AI-treated patients (Supplementary Fig. 9c)[33,34].

**KRT80 drives cytoskeletal changes involved in migration and invasion.** To test if KRT80 manipulation drives ancillary

phenotypes synergistic to cytoskeletal changes, we performed RNA-seq in cells transfected with KRT80 but where SREBP1 is not yet activated (non-invasive MCF7 cells, Fig. 7a). Ectopic KRT80 expression led to clear transcriptional differences dominated by the reprogramming of a small set of genes (Fig. 7a, b and Supplementary Data 2). Pathway analyses of upregulated genes pointed out to cytoskeletal rearrangements (Fig. 7c and

**Fig. 5** KRT80 directly promotes cell invasion. **a** Design of the 3D invasion assay. Organoids were derived from treatment naive (green; MCF7) or invasive AI resistant (orange; LTED) breast cancer cells. KRT80 expression was manipulated via ectopic overexpression or sh-mediated stable depletion. Organoids were embedded in Matrigel and monitored for 48 h. **b** Representative brightfield images of KRT80-manipulated organoids. Panels show results obtained in KRT80 depleted cells. **c** Representative brightfield images of KRT80-manipulated organoids. Panels show results obtained in KRT80 over-expressing cells (DKK-tagged KRT80). Small inset number represent normalized fold area changes of each represented experiment. Bars scale = 400 μm. **d** Quantification of the area fold change in organoids overexpressing KRT80 or KRT80 knock-down LTED cells in 3D invasion assay normalized to MCF7 (*$p < 0.05$, **$p < 0.01$, Student $t$ test; $n = 3$ biological triplicates in which at least 4 organoids were measured). Data is presented as mean ± SD. **e** Confocal microscopy of matrigel embedded invasive AI resistant LTED organoids. **f** Replication dependent labeling of breast cancer spheroids. Cells were labeled with CMFDA that is converted to its membrane-impermeant fluorescent form by cytosolic esterase to entrap the dye. Active replication can dilute the dye until disappearance within 2–3 cell cycles. **g** Quantification of the area fold change in organoids treated with CMFDA. Lines represent mean and SD. Asterisks represent significance level $p < 0.05$ after Student $t$ test. **h** Representative images of CMFDA tagged spheroids. Invasive borders are highlighted by dotted white lines. Representative original borders are highlighted by yellow dotted lines. Bars scale = 400 μm

Supplementary Data 2). Amongst them, we found particularly striking the strong KRT80-dependent induction of cortactin (*CTTN*), a factor directly linked to actin rearrangements, lamellipodia formation and cancer cell invasion[34,35], that we confirmed by immunofluorescence (Fig. 7d). In addition, we also detected a significant upregulation of *SEPT9*, a member of the septin family directly linked to actin fiber formation, focal adhesion maturation, and motility[36,37] (Fig. 7b). Genes activated in response to KRT80 upregulation have prognostic value, even when other classical clinical features are considered (Fig. 7e). These data parallel KRT80 prognostic features and hint that these genes might underlie early metastatic invasion (Fig. 7e). We also observed that several genes negatively regulated by KRT80 induction play central roles in cancer biology including negative regulators of migration (PCDH10, CADM1), tumor suppressors such as CDKN1A and PDCD2, genes involved in DNA repair (RAD50), chromatin remodelers as SMARCE1 and CHD4 and tumor specific antigens (CD276) suggesting a direct link between cytoskeletal reprogramming and several other oncogenic phenotypes (Fig. 7b). Together, these results further support that KRT80 manipulation is sufficient to activate genes driving dramatic cytoskeletal rearrangements that ultimately induce invasive behaviors in BC and poorer prognosis. We cannot speculate at the moment if this is driven by a cytoskeleton-transcriptional feedback or it is mediated by some specific transcriptional factors.

## Discussion

The relationship between drug-resistance and phenotypic reprogramming in breast cancer has not been studied in detail, as generally the focus has been on characterizing the mechanisms of resistance rather than the associated changes in traits that might possibly play a role in shifting cancer cell behaviors. Furthermore, it is known that aberrant cytoskeletal architecture characterizes tumor cells and it is associated with cell migration and invasion; yet the endogenous and exogenous triggers underlying cytoskeletal reorganization in tumor cells are not well understood. Here, we have uncovered a novel and causal link between endocrine therapy resistance, intra-tumoral stiffness and augmented invasive potential in luminal BC (Fig. 7f). Our data strongly suggest that therapy plays a direct role in shaping the biophysical properties and invasive potential of cancer cells, by inducing epigenetic rearrangements leading to KRT80 upregulation and concomitant cytoskeletal reorganization. Our data strongly suggest that SREBP1 is the link between drug-resistance and cytoskeletal reprogramming. Upon long-term AI treatment, SREBP1 mediates the activation of pro-survival pathways[7] by promoting the cell-autonomous production of endogenous ERα ligands. In addition, SREBP1 is also recruited at the KRT80 enhancer, a non-canonical SREBP1 target, leading to KRT80 transcription in drug-treated cells. This mechanism does not appear to be promoted by absolute changes in SREBP1 abundance, but rather by enhanced

chromatin binding. Furthermore, it is important to note that our data demonstrate that SREBP1 is essential but might not be sufficient for KRT80 activation. How SREBP1 is capable to sense AI-mediated stress needs to be worked out mechanistically, but overall these data support SREBP1 as a potential target to antagonize BC progression. We also describe an unexpected role for intermediate filaments in promoting cancer cell invasion by showing for the first time that KRT80 promotes actin cytoskeleton rearrangements. These are characterized primarily by the formation of lamellipodia and mature focal adhesions, which are critical structures required for migration in complex environments[33]. Our data might also reconcile some previous observations that were in an apparent contrast. Few clinical studies have highlighted that stiffer BC lesions do carry worse prognosis[27,29–31], while others suggested that EMT-like processes, necessarily decrease intracellular stiffness, are needed for tumor progression. The link between treatment, KRT80 activation and increased stiffness would fit with several of these observations, especially in the light of collective-invasion phenotypes observed for ERα-positive BC cells[27,28]. Larger longitudinal clinical studies measuring stiffness and KRT80 activation in endocrine neoadjuvant-treated patients are needed and should be linked to long-term monitoring for distal relapse.

A directional link between epigenetic and cytoskeleton reprogramming was not described before and it offers an intriguing axis for drug development and biomarker discovery, especially within the goal of preventing metastatic invasion in BC patients treated with aromatase inhibitors.

## Methods

**Cell lines and cell culture**. All cell lines used in the study were karyotyped and validated and no cell lines from the ICLAC database were used. In this study we used MCF7 breast adenocarcinoma cell line and derived resistant clones (Supplementary Fig. 1). MCF7 Tamoxifen Resistant cell line (MCF7T) was derived from MCF7 upon one-year treatment with Tamoxifen. MCF7 Fulvestrant Resistant cell line (MCF7F) was derived from MCF7 upon one-year treatment with Fulvestrant. LTED (Long Term Estrogen Deprivation) cell lines were derived from MCF7 cell line upon one-year estrogen deprivation, mimicking aromatase inhibitor resistance. LTED Tamoxifen Resistant cells (LTEDT) were derived from LTEDs upon one-year Tamoxifen treatment. LTED Fulvestrant Resistant cells (LTEDF) were derived from LTEDs upon one-year Fulvestrant treatment[38,39]. In addition, we employed an alternative aromatase inhibitor resistant model: T47D breast adenocarcinoma cell line and T47D-LTED. The latter was derived from T47D parental upon six months of estrogen deprivation. MCF7 and T47D breast cancer cell lines were cultured in DMEM (Sigma-Aldrich) supplemented with 10% FCS (Fetal Calf Serum, First Link UK), 2 mM L-Glutamine, 100 units/mL penicillin, and 0.1 mg/mL streptomycin (Sigma). MCF7 were further supplemented with 10-8 M Estradiol (Sigma). Estrogen-deprived cell lines (LTED, LTEDT, and T47D-LTED) were cultured in phenol-red free DMEM (Gibco, Life Technologies) supplemented with 10% DC-FCS (Double Charcoal stripped Fetal Calf Serum, First Link UK) and 2 mM L-Glutamine, 100 units/mL penicillin, and 0.1 mg/mL streptomycin (Sigma-Aldrich). LTEDT were further supplemented with 10–7 M Tamoxifen (SIGMA). All cells were tested for mycoplasma contamination using MycoAlert mycoplasma detection kit Assay Control Set by Lonza (LT07-518) following manufacturer's instructions.

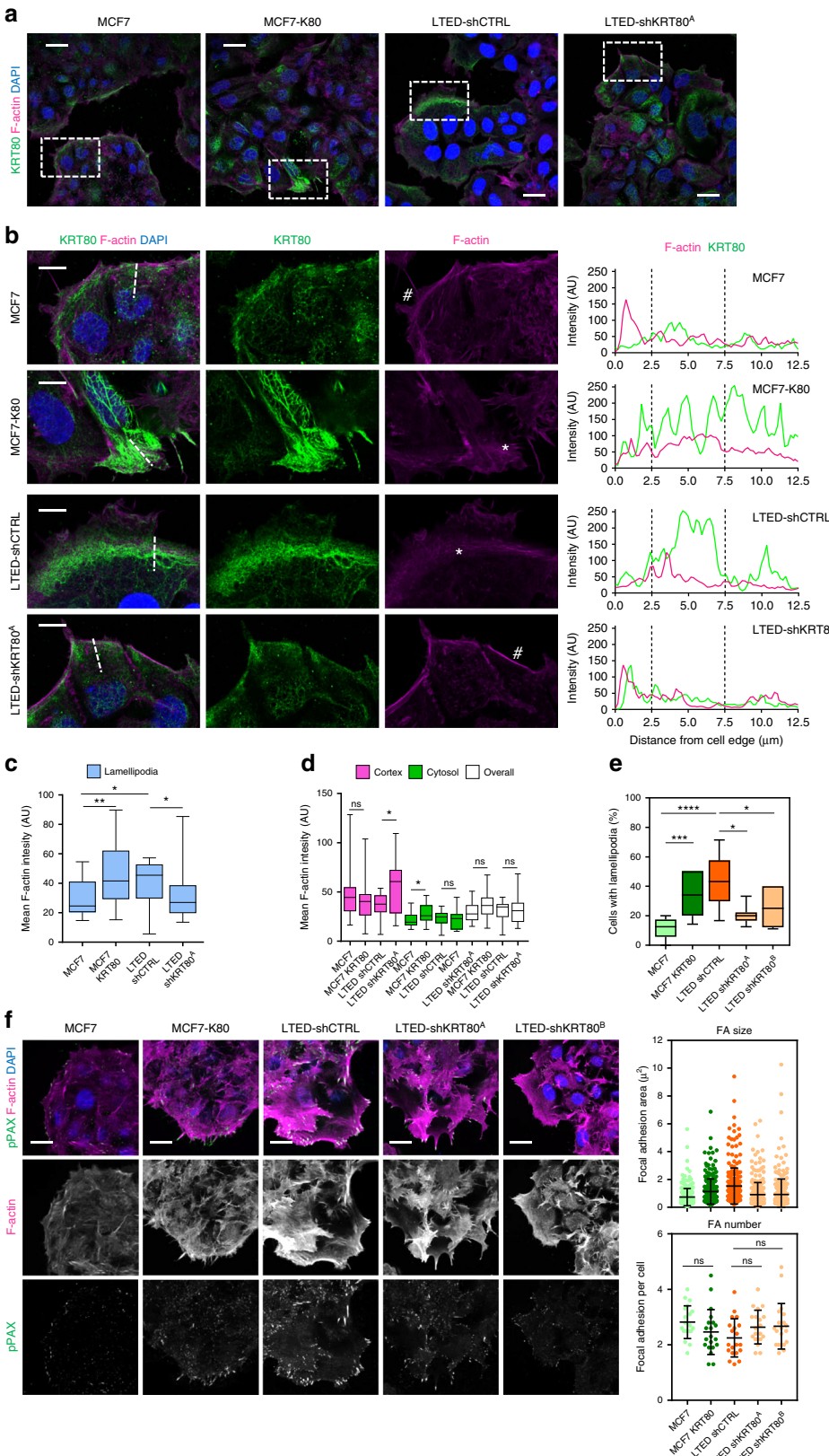

**Generation of stable cell lines.** For KRT80 overexpression, a full length KRT80 cDNA clone Myc-DKK-tagged was obtained from OriGene and transformed into DH5α competent cells (Invitrogen). Plasmid DNA was isolated using Maxi-Prep Kit (QIAGEN) and transfected in MCF7 and LTED cells using X-tremeGENE 9 DNA Transfection Reagent (Roche) following manufacturer's instructions. Transfected cells, carrying Neomycin resistance, were selected with G418 (SIGMA), used at a final concentration of 1 mg/mL for MCF7 and 0.5 mg/mL for LTED.

Knock-down of KRT80 was achieved by transfection of two different shRNA expression vectors and a scrambled negative control obtained from OriGene. Cells carrying the corresponding construct were selected with Puromycin (Sigma-Aldrich) at a final concentration of 1 ug/mL for MCF7 and 0.5 ug/mL for LTED cell line. NucLight Red Lentivirus (IncuCyte, 4627) was used to infect MCF7 and generate MCF7 mKate2. Stable and polyclonal cell populations were established after Zeocin selection (300 μg/ml).

**Fig. 6** KRT80 induces invasion-associated cytoskeletal changes. **a** Representative confocal microscopy images showing F-actin (magenta), KRT80 (green) and DAPI (blue) staining of MCF7-control, MCF7-K80, LTED-control and LTED-sh[a] cells. Scale bars represent 25 µm. **b** Zoom-up magnifications of areas indicated in **a**, showing F-actin (magenta), KRT80 (green) and DAPI (blue) staining in cells located at the border of clusters. Single channel images for F-actin and KRT80 are also shown. Scale bars, 10 µm. Asterisks indicate lamellipodia-like structures in MCF7-K80 and LTED cells, and hashtags indicate cortical actin areas in MCF7 and LTED-sh[a] cells. Graphs on the right show line scan analysis for F-actin and KRT80 fluorescence across the leading edges of cells, as indicated in the broken line in the merged images. **c, d** Graphs show quantification of F-actin fluorescence intensity at lamellipodial regions (**c**) and at cell cortex, cytosol and overall (i.e., whole cell) (**d**) in MCF7-control, MCF7-K80, LTED-control and LTED-sh[a] cells ($n = 19$, MCF7; $n = 20$, MCF7-K80; n = 14, LTED; $n = 16$, LTED-sh[a] individual cells). **e** Graph shows quantification of percentage of cells with clear lamellipodia and membrane ruffles in MCF7-control, MCF7-K80, LTED-control, LTED-sh[a] and LTED-sh[b] cells ($n = 8$, MCF7; $n = 12$, MCF7-K80; $n = 12$, LTED; $n = 7$, LTED-sh[a]; $n = 6$, LTED-sh[b] fields of view). **f** Representative confocal microscopy images showing F-actin (magenta), pY118-Paxillin (green) and DAPI (blue) staining of MCF7-control, MCF7-K80, LTED-control, LTED-sh[a] and LTED-sh[b] cell. Scale bars, 25 µm. Graphs show quantification of individual leading-edge focal adhesion size (left) and number of adhesions per cell (right). Focal adhesion size ($n = 269$, MCF7; $n = 251$, MCF7-K80; $n = 257$, LTED; $n = 331$, LTED-sh[a]; n = 276, LTED-sh[b]). Focal adhesion number ($n = 20$, MCF7; $n = 20$, MCF7-K80; $n = 20$, LTED; $n = 20$, LTED-sh[a]; $n = 20$, LTED-sh[b], individual cells). Statistical analyses were performed using one-way ANOVA with Tukey's post-test. Floating bars and lines represent mean, inter-quantile distribution and SD. Asterisks represent significance at $*p < 0.05$, $**p < 0.01$ and $***p < 0.001$ levels

**Live cell imaging and data analysis**. Live cell imaging was performed on Incu-Cyte ZOOM (Essen BioScience) equipped with temperature, humidity and $CO_2$ control. Images were acquired every 6 h with 10× plan fluorescence objectives for the proliferation assay. Data were analyzed and plotted using Prism6. Individual cells were counted longitudinally to verify absence/presence of proliferation/cell death.

**TAD analysis**. TADs were identified using Hi-C data from IMR90 and H1 stem cells as described in ref. [7] (http://chromosome.sdsc.edu/mouse/hi-c/download.html). Acetylation profiles were averaged on each TAD locus and difference in normalized read numbers between TAD loci from MCF7 cells or ET-treated cells were calculated. Difference were expressed in terms of positive or negative ratios and ranked according to increase or decrease acetylation[7]. The Type II Keratin Locus was identified within the top 5% of TAD which increase acetylation during the acquisition of ET-resistance (comparing MCF7 with LTED cells[7]). In the current manuscript we have used a similar strategy while comparing MCF7 with all ET derivatives. Actual averaged read number/TAD are now plotted in Fig. 1a according to each cell line.

**RNA extraction and RT-qPCR**. Cells were washed with PBS and harvested using a cell lifter (Corning) in RLT buffer supplemented with 1% β-mercaptoethanol. Cell lysate was homogenized using QIAshredder columns (QIAGEN) and RNA extraction was performed with RNeasy Mini Kit (QIAGEN) following manufacturer´s instructions. RNA concentration was measured using a NanoDrop 1000 Spectrophotometer and 0.5–2 µg of RNA were retrotranscribed using High Capacity cDNA Reverse Transcription Kit (Applied Biosystems). Quantitative PCR (qPCR) was performed using 2× SYBR GREEN Mix (Invitrogen) and expression levels of each gene were calculated using the 2-ΔΔCt method, normalizing expression levels to 28S transcript.

**Protein extraction, quantification, and western blotting**. Cells were harvested in 50 µL ice-cold RIPA buffer (50 mM Tris- HCl at pH 8.0, with 150 mM sodium chloride, 1.0% Igepal CA-630 (NP-40), 0.5% sodium deoxycholate, and 0.1% sodium dodecyl sulfate) (Sigma; #R02780), supplemented with 1× protease (Roche; #11697498001) and 1× phosphatase (Sigma; #93482) inhibitor cocktail. The cell pellet and RIPA were mixed by pipetting up and down, incubated at 4 ºC for 30 min and vortexed every 5 min. Cell lysates were then centrifuged at 13,000 rpm for 30 min at 4 ºC. The supernatants were transferred to a new 1.5 mL eppendorf tube and the pellets were discarded. Protein concentration was measured using BCA Assay Kit (Thermo Fisher) following manufacturer's instructions. With regard to western blotting, 20 µg of protein per sample, were mixed with 4× Bolt sample buffer (Life Technologies; #B0007), 10× Bolt sample reducing agent (Life Technologies; #B0009), ddH2O and heated at 95 ºC prior to loading. Protein lysate were loaded into BOLT 4-12% Bis-Tris Plus Gel (Life Technologies; NW04120BOX). The pre-made gel was placed into a mini gel tank (Life Technologies; #A25977) containing 1× Bolt running buffer (Life Technologies). Electrophoresis was carried out at 90 V for 35 min to allow proteins to adequately run through and also until the bromophenol blue dye reached the bottom of the gels. The gels were transferred into a Biotrace nitrocellulose membrane (VWR; #PN66485) using a TE-22 transfer unit (Hoefer GE Healthcare) at 100 V for 90 min. The membrane was incubated in blocking buffer for 45 min at room temperature to reduce non-specific binding of primary antibody. The membrane was then incubated with the diluted primary antibodies (Anti-KRT80 for cell lines characterization from[12], Anti-KRT80 for shRNA and IHC from HPA 077836 and 077918, Atlas Antibodies (1:200 dilution), Anti-SREBP1 H-160 sc-8984 Santa Cruz Biotechnology (1:200 dilution), (Guinea Pig Anti-KRT80 1:5,000; Mouse Anti-DKK 1:1,000, OriGene; Mouse Anti-β-Actin 1:10,000) in blocking buffer at 4 ºC and allowed to shake overnight. After primary antibody incubation, the membrane was washed three times in PBST (5 min per

wash on a rocking platform) and then incubated for 1 h with the HRP-GAPDH (Abcam; #ab9482 (1:5000 dilution)) conjugated antibody (for the loading control membrane) which was diluted in 5% BSA/PBST and goat anti-rabbit IgG (H + L) Cross Absorbed secondary antibody, HRP 1:20000 dilution (ThermoFisher Scientific; #31462). The membranes (including the loading control membrane) were washed three times in PBST. Amersham ECL start Western Blotting Detection reagent (GE Healthcare Life Sciences; #RPN3243) was used for chemiluminescent imaging using the Fusion solo (Vilber; Germany) imager.

**Chromatin immunoprecipitation (ChIP)**. For ChIP, cells were fixed with 1% formaldehyde for 10 min at 37 °C and reaction was quenched with 0.1 M glycine. The cells were subsequently washed twice with PBS after which they were lysed in lysis buffer (LB) 1 (50 mM HEPES-KOH, pH 7.5, 40 mM NaCL, 1 mM EDTA, 10% glycerol, 0.5% NP-40, 0.15% Triton X-100), for 10 min, then for 5 min in LB 2 (10 mM Tris-HCl, pH 8.0, 200 mM NaCl, 1 mM EDTA and 0.5 mM EGTA) and subsequently eluted in LB 3 for sonication (10 mM TRIS-HCl, pH 8.0, 100 mM NaCl, 1 mM EDTA, 0.5 mM EGTA, 0.1% Na-Deoxycholate, and 0.5%N-lauroylsarcosine). DNA was sheared using the Bioruptor® Pico sonication device (High, 10 cycles of 30" on and 30" off) (Diagenode). Sheared chromatin was cleared by centrifugation. Magnetic beads were precoated by adding 10 µg of antibody Rabbit-anti-SREBP1 (H-160): sc-8984 (Santa Cruz Biotechnology, Inc.); Rabbit-anti-Histone H3 acetyl K27 antibody (abcam, ab4729); Rabbit- anti-Histone H3 (monomethyl K4) antibody (abcam, ab8895); Rabbit- anti-Histone H3 (dimethyl K4) antibody (abcam, ab7766) to 50 µl magnetic beads per ChIP (Dynabeads protein A, Life technologies) and incubated for 6 h on a rotating platform at 4 °C. Diluted sheared chromatin was added to the coated magnetic beads and incubated on a rotating platform at 4 °C O/N. Ten microliter of sheared chromatin taken as input and treated the same. The next day magnetic bead complexes were washed three times with RIPA buffer (50 mM HEPES pH 7.6, 1 mM EDTA, 0.7% Na deoxycholate, 1% NP-40, 0.5 M LiCL) and two times with TE buffer (10 mM Tris pH 8.0, 1 mM EDTA). DNA is O/N eluted from the beads in 100 µl de-crosslinking buffer (50 mM Tris-HCl, pH 8.0, 10 mM EDTA, 1% SDS) at 65 °C. After overnight de-crosslinking, DNA was treated with 2.7 µl of 1 mg/ml RibonucleaseA (RNaseA) for 30 min at 37 °C and subsequently incubated with 1.3 µl of 20 mg/ml proteinase K (Invitrogen) for 1 h at 55 °C. Then DNA extraction was performed using SPRI magnetic beads (Beckman Coulter, B23318). After elution in TE buffer, DNA was quantified using Qubit (ThermoFisher Scientific; Qubit 3.0 Fluorometer; #Q33216) high sensitivity assay (ThermoFisher Scientific; #33216). Quantitative polymerase chain reaction (qPCR) was then carried out (Applied Biosystems; #7900HT Real time PCR, #StePOnePlus). If sufficient enrichment is seen in the antibody treatment samples over the 'input' samples and compared with internal negative controls, these undergo DNA size selection and library preparation.

**Library preparation and ChIP-seq data analysis**. Prior to sequencing, ChIP samples were library prepared using the NEBNext Ultra II DNA Library Prep Kit for Illumina (New England Biolabs, NEBNext Ultra II DNA library prep kit for Illumina, #E7770, NEBNext Multiplex Oligos for Illumina, #E7335L). Adaptor ligated DNA was size selected with SPRI magnetic beads (Beckman Coulter, B23318) which aims to retain DNA fragments between 200–300 base pairs (bp), recognizable for the Illumina sequencer (#NextSeq500). After library preparation, we performed qPCR, high sensitivity DNA quantification and size selection measurement (Agilent Bioanalyzer 2100 system + High sensitivity DNA measurement assay; 5067–4626) before sending samples for sequencing. Raw sequencing files processed by the Illumina NextSeq500 sequencer were obtained in "FASTQ" format. The raw sequencing files were then aligned to the genome using Bowtie 1.11 short reads sequence aligner using the human reference genome 19 (Hg19) as the reference genome. The output of Bowtie 1.1.1 is the "SAM" file extension format, for both input (control) and ChIP samples, which were then used by

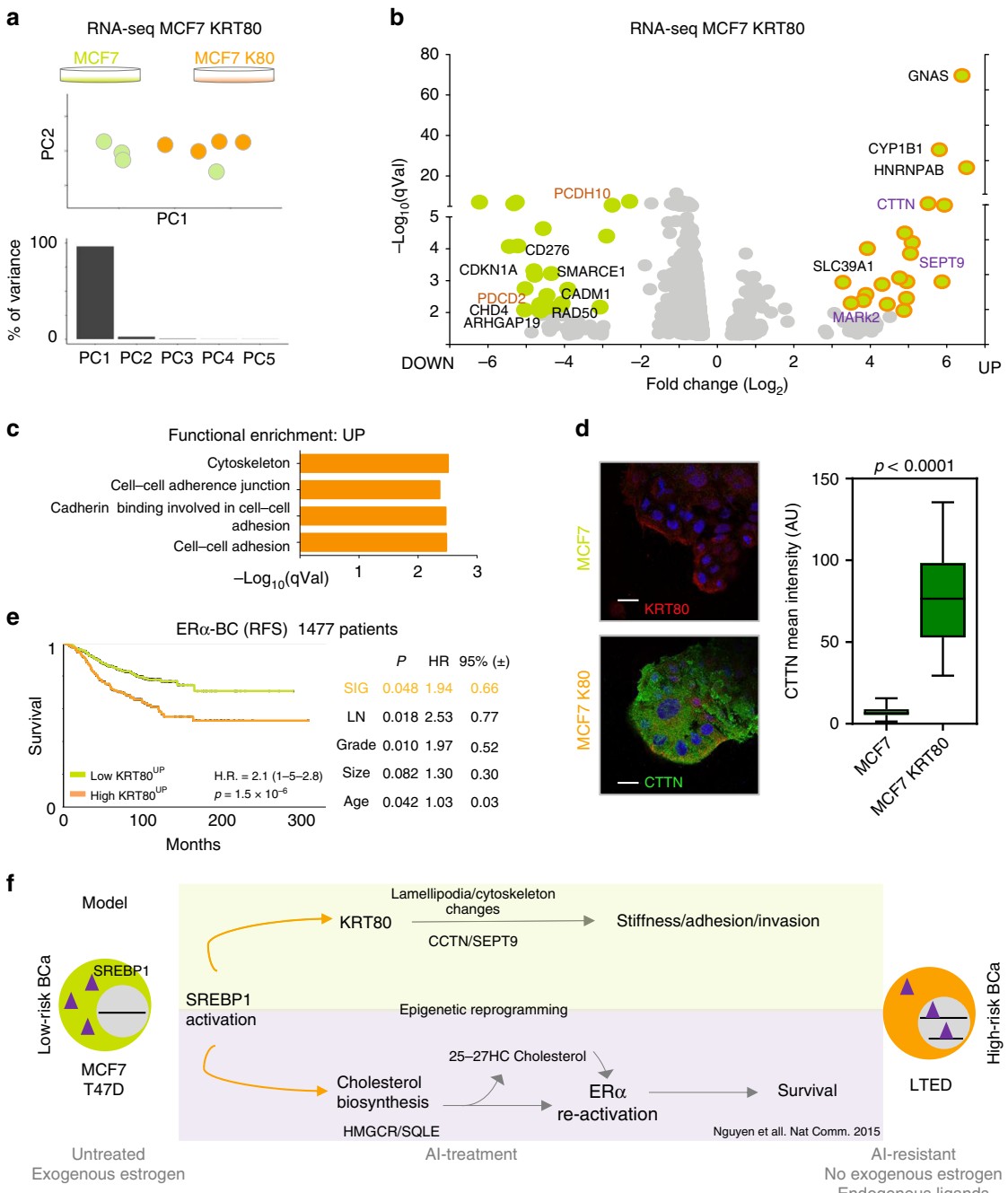

**Fig. 7** KRT80-changes induce transcriptional changes of cytoskeletal genes. **a** PCA analyses of RNA-seq profiled MCF7 breast cancer cells or MCF7 cells with ectopic expression of KRT80. **b** Volcano plots of over-expressed or under-expressed genes in MCF7 cells following KRT80 ectopic expression. For a complete list, see Supplementary Data 2. **c** Functional enrichment for upregulated genes following KRT80 ectopic expression. **d** Representative confocal microscopy images showing F-actin (magenta), cortactin (CTTN, green) and DAPI (blue) staining of MCF7-control and MCF7-K80 cells. Scale bars represent 25 μm. Graph shows mean fluorescence intensity of cortactin in MCF7-control and MCF7-K80 cells ($n = 40$, MCF7; $n = 4$, MCF7-K80 individual cells). **e** Kaplan-Meier plot of ERα-positive breast cancer patients dichotomized to average high or low expression for genes upregulated in response to KRT80 over-expression (Panel **b**). Multivariate statistics are shown on the right inside table. **f** Current model: long-term AI treatment promotes constitutive activation of SREBP1 leading to pro-survival re-activation of estrogen receptor[12], and global cytoskeletal re-arrangements. Cytoskeletal re-organization leads to direct biomechanical changes and promotes pro-invasive behavior

Model-based analysis for ChIPSeq (MACS) version 1.42 for peak calling; with all peaks called at a Q-value cut-off of $10^{-3}$ and default settings applied. MACS outputs result in "BED" file format and "WIG" files.

**Digital footprinting**. Data from digital footprint were obtained from[7]. Briefly, DHS-seq libraries were analyzed using Wellington[40] with the following parameters: -fdr 0.01 -pv "$-5, -10, -20, -30, -50, -100$", using the DHS called using MACS v1.4 with a threshold of $1e-10$.

**RNA sequencing and single cell RNA-seq**. Total RNA from each sample was quantified by Qubit® Fluorometer and quality checked by Agilent Bioanalyzer® RNA 6000 Nano Chip. All samples have high quality RNA with a RIN score > 7. One microgram of total RNA from each sample was used as starting material for paired-end RNA-seq library preparation using NEBNext rRNA Depletion Kit (NEB #E6310) and NEBNext Ultra II RNA Library Prep Kit for Illumina (NEB #E7770) following the manufacturer's instructions. Libraries were sequenced on an

Illumina Next Seq machine (#NextSeq500). Reads were processed using Kallisto and DEGS were called using Sleuth[41].

For single cell RNA-seq analyses, only cells showing at least 5000 detected transcripts were considered. Single-cell experiments were performed as described[38]. Briefly, cells were processed using 10× genomics platform (v2.3 kits). Barcodes were demultiplexed using 10× internal pipeline. Expression profiles from MCF7 cells either from red media ($n = 1227$) or two days of estrogen-deprivation ($n = 1193$) were then normalized using the R package Scran (v1.6.9)[42]. Differential expression between the two conditions was estimated using the Two-sample Likelihood Ratio Test implemented in the LRT function of the MAST R package (v1.4.1)[43].

**Single cell RNA-FISH.** Cell were cultured, fixed and pretreated according to the protocol for the RNAscope® Multiplex Fluorescent Reagent Kit v2 Assay provided by Advanced Cell Diagnostics (ACD, #323100, Nunc Lab Tek II 2 Well Glass Slides, #154461K). The probes were as follows: RNAscope® 3-plex Positive Control Probe (320861), RNAscope® Negative Control Probe – Bacillus subtilis dihydrodipicolinate reductase (dapB) gene (310043), RNAscope® Target Probe C1 (20ZZ probe named Hs-KRT80 targeting 294-1445 of NM_182507.2, 300031), RNAscope® Target Probe C2 (RNAscope® Probe - Hs-SQLE 465071, 300031), RNAscope® Target Probe C2 (20ZZ probe named Hs-SREBF1 targeting 958-2002 of NM_001005291.2, 300031) and RNAscope® Probe - Hs-HMGCR (RNAscope® Target Probes, 470561). The assay was run following the manufacturer's instructions, hybridization was performed overnight. PerkinElmer TSA Plus Fluorophores (fluorescein, NEL741001KT and Cyanin 3, NEL744001KT) were diluted at 1:1300 and assigned to the channels HRP-C1 and HRP-C2, respectively. Fluorophores: PerkinElmer TSA Plus Fluorescein System (NEL741001KT) and PerkinElmer TSA Plus Cyanine 3 System (NEL744001KT). Samples were imaged using a ×60 objective with a Ti Nikon microscope equipped with a spinning disk (CAIRN) and analysed in Image J.

**3D Organoid assay.** A total of 250,000 cells were resuspended in 1 mL of the corresponding media and 20 μL drops were placed in the lid of a 10 cm dish (Corning). The lid was flipped over the dish containing 5 mL of media in order to prevent evaporation. Hanging drops were incubated for 5 days at 37% C in a humidified atmosphere, during which formation of organoids was achieved. Before being included in 3D matrix for the invasion assay, the organoids were collected and labeled with 10 μM CellTracker™ Green CMFDA (Thermo Fisher, Waltham, USA) dye by incubating them in serum free media for 45 min at 5% $CO_2$. Labeling solution was removed, and spheroids were washed in cell medium. To follow, spheroids were centrifuged at 300 rpm, immersed in 10 μL of phenol-red free Matrigel® (BD Biosciences) and placed in a 24 well-plate (Corning) The appropriate media containing G418 or puromycin was subsequently added to the well. Brightfield images were acquired at days zero and day two using an EVOS microscope (Advanced Microscopy Group, Life Technologies). Images were analyzed using Fiji ImageJ software and fold-change area was calculated using the following formula: Area (fold-change) = Area Day 2/Area Day 0.

**Immunofluorescence and confocal microscopy.** Organoids were washed with PBS and fixed for 15 min with 4% PFA/PBS. Fixation was stopped by rinsing with 100 mM Glycine/PBS. Cells were permeabilized with 0.5% Triton/PBS X-100 and unspecific binding was blocked with blocking solution (5% BSA, 0.2% Triton X-100, 0.05% Tween in PBS) for 90 min. Organoids were then incubated with primary antibody (Rabbit Anti-KRT80 1:200, Sigma-Aldrich) for 2 h, washed three times with washing buffer (0.2% Triton X-100, 0.1% BSA, 0.05% Tween in PBS), and incubated with secondary antibody (Goat Anti-Rabbit Alexa Fluor 555 1:200, Invitrogen) for 45 min. Organoids were washed with immunofluorescence buffer for 20 min and PBS for 10 min. Finally, organoids were mounted in Moviol (AppliChem) containing 5 μg/mL of DAPI (Lonza) and visualized using a Zeiss LSM-780 inverted confocal microscope.

**Immunofluorescence.** Cells were seeded on glass bottom 24 well plates (MatTek) coated with 10 μg/ml fibronectin (Sigma), fixed in 4% PFA and permeabilized in PBS with 0.2% Triton X. The samples were blocked in 3% BSA with 0.1% PBS Tween (PBST) for 3 h. The primary antibodies (Cortactin, 05-180, Millipore, 1:100; Keratin-80, HPA077836, Sigma Atlas, 1:100; Phospho-Paxillin-pY118, 44-722 g, Invitrogen, 1:100) were diluted in 3% BSA in 0.1% PBS Tween and incubated overnight at 4 °C. The wells were then washed 3 times in 3% BSA 0.1% PBST for 10 min, followed by the addition of the appropriate secondary antibody (Alexa Fluor, Invitrogen), DAPI (Sigma) and FITC-phalloidin (Sigma). Imaging was performed using Leica SP8 Confocal microscope.

**Image analyses.** Cells stained for KRT80 and F-actin (Phalloidin) were imaged with a ×63 oil immersion objective. Cells were assessed for lamellipodia formation based on morphology and formation of lamellipodial structures. Only cells at the border of clusters were evaluated. Cells were positive if a clear membrane ruffle and lamellipodia towards the leading edge (i.e., free space) was observed. Values represent the percentage of positive cells per field of view. Analysis of F-actin and KRT80 fluorescence intensity was performed in confocal images acquired at the same time at identical laser settings. Analyses of F-actin at different cell regions were performed using Image J, analyzing 2–3 representative cells at the border of clusters per image. Areas at the cell cortex, lamellipodia and cytosol were delineated using the free-hand drawing function and area and mean F-actin fluorescence intensity measured. To calculate the overall (i.e., whole cell) fluorescence intensity, the total intensity of cortical, lamellipodial and cytosolic F-actin was calculated and divided by the total area analyzed. Line scan analyses were generated using the line intensity function in Leica's Application Suite X software. The fluorescence intensity of F-actin and KRT80 as a function of the distance from the cell edge was obtained from confocal images acquired at the same time at identical laser settings. Lines (12.5 μm) used for the analysis are indicated in the respective figures. Values correspond to the relative fluorescence intensity for each staining.

For analysis of pY118-paxillin adhesion size, cells were imaged using a ×63 oil immersion objective and analyzed using Volocity (Perkin Elmer). Only cells at the border of clusters (leading edge) were analyzed. Individual pY118-Paxillin adhesions towards the leading edge were identified, selected using the magnetic lasso tool and the size measured using Volocity. Values represent the mean FA size in μm² per cell. For quantification of focal adhesion number, individual cells were identified and the number of pY118-paxillin adhesions at the leading edge per cell was manually quantified.

To determine the mean cortactin (CTTN) fluorescence intensity, cells were imaged using a ×63 oil immersion objective at the basal plane. Individual cells were identified, selected and the mean fluorescence intensity per cell was determined using Volocity. Values correspond to the mean CTTN fluorescence intensity per cell for each staining.

**Tissue specimens.** Seventy-five human breast specimens and ten metastatic lymph nodes were selected from Histopathology Department at Charing Cross Hospital, with the previous approval of Imperial College Healthcare NHS Trust Tissue Bank.

A Tissue Microarray (TMA) containing 26 primary breast tumors and paired ETR relapses was constructed as previously described[18].

Immunohistochemistry staining was scored using a quick score system by two independent investigators, one of them a consultant pathologist (SS). Score was calculated as follows: S = 3 (strongly stained cells), S = 2 (moderate staining), S = 1 (poorly stained cells), and S = 0 (absence of staining). Staining intensity was assessed as mean intensity from the tumor region contained within the TMA. A second set of tissues (pre and post-adjuvant therapy) was constructed at the Istituto Nazionale Tumori (Milan) with material from the INT Tissue Bank. All specimens were obtained from consented-patients (Imperial College NHS and INT tissue banks).

**Immunohistochemistry.** Formalin fixed and paraffin embedded (FFPE) tissue specimens were sliced in 4 μm sections using a Leica RM2235 manual microtome. Dried sections were de-waxed by immersion in xylene and rehydrated with subsequent immersion in 100% ethanol, 70 % ethanol and distilled water. Antigen retrieval was performed by immersion in PBS 0.01 M citric acid pH 6 and heated at 800 W for 15 min. Slides were rinsed in PBS and endogenous peroxidase activity was blocked for 30 min using Dako RealTM Peroxidase Blocking Solution. Following that, slides were rinsed twice with PBS and incubated with 10% pig serum (Bio-Rad) for 30 min and overnight with KRT80 antibody (Sigma-Aldrich, 1:200). Following day, slides were rinsed in PBS and incubated 30 min with secondary antibody (biotinylated Goat Anti-Rabbit IgG 1:200, Vector Laboratories) and 30 min with an avidin/biotin peroxidase-based system (VECTASTAIN Elite ABC Kit, Vector Laboratories). Color reaction was developed for 1 min using DAB (Diaminobenzidine, Vector ImmPACT DAB Peroxidase Substrate). Color development was stopped by immersion during 5 min in running tap water and following that, nuclei was stained with haematoxylin. Slides were dehydrated in 100% ethanol, cleared in xylene and mounted in DPX (SIGMA).

**Statistical analysis.** Data is presented as mean ± SD (standard deviation) in most figures. Whenever this is not the case, the figure legends states the exact details. Data analysis was performed using GraphPad Prism 6 software. Statistics are described in details in each figure legend. Generally, Student $t$ test and one-way ANOVA were applied. The sue of additional statistical methods, such as nonparametric Mann–Whitney test, are described in individual figure legends.

**Survival analysis.** Publicly available breast cancer datasets were identified in GEO (https://www.ncbi.nlm.nih.gov/geo/), EGA (https://www.ebi.ac.uk/ega/home), and TCGA (https://cancergenome.nih.gov/). Only cohorts including at least 30 patients and with available follow-up data were included. Samples derived using different technological platforms (Affymetrix gene chips, Illumina gene chips, RNA-seq) were processed independently. For KRT80, the probe set 231849_at was used in the Affymetrix dataset, the probe ILMN_1705814 was used in the Illumina dataset and the gene 144501 was used in the RNA-seq dataset. Cox proportional hazards survival analysis was performed as described previously[44]. Kaplan–Meier plots were derived to visualize survival differences. In the multivariate analysis, the RNA expression of ERα, HER2, and MKI67 were used as surrogate markers for ER and HER2 status, and for proliferation. In this, the probe sets 205225_at, 216836_s_at,

and 212021_s_at were used for ERα, HER2, and MKI67, respectively. The survival analysis was performed for relapse-free survival (RFS), overall survival (OS), and post-progression survival (PPS). PPS was computed by extracting the RFS time from the OS time for patients having both RFS and OS data and having an event for RFS. Censoring data for PPS was derived from the OS event. The survival analysis was performed in the R statistical environment.

**Cellular microrheology.** To characterize the mechanical properties of the four different BC cell lines, we used magnetic tweezer microrheology to measure cell deformation in response to magnetically generated forces. Tensional magnetic forces were induced by a high gradient magnetic field generated by an electromagnetic tweezer device. The positioning of the tip of the magnetic tweezer device was controlled by an electronic micromanipulator. Superparamagnetic 4.5 μm epoxylated beads (Dynabeads, Life Technologies) were coated with fibronectin (40 μg per $8 \times 10^7$ beads, Sigma Aldrich F0895) and incubated with adherent cells for 30 min, prior to measurements, to allow integrin binding and provide a mechanical link between the bead and the cytoskeleton. The unbound beads were removed by multiple washing with PBS. The experiments were performed at 37 °C, 5% $CO_2$ and 95% humidity in DMEM containing 2% FBS in a microscope stage incubation chamber. A viscoelastic creep experiment was conducted by applying mechanical tension onto single beads bound on the apical surface of the cells with a constant pulling force ($F_0 = 1$ nN) for 3 s generated by the magnetic tweezers. The viscoelastic creep response of the cells was recorded by tracking the resulting bead displacement in brightfield (×40 objective at 20 frames per second, Nikon Eclipse Ti-B) that is indicative of the local cytoskeletal deformation. A custom-built MATLAB algorithm was then used to analyze the image sequences and track bead displacement by following the intensity-weighted centroid of the bead across all captured frames. The viscoelastic creep response $J(t)$ of cells during force application followed a power-law in time $J(t) = J_0(t/t_0)^\beta$ with the prefactor $J_0$ representing cell compliance ($J_0$ = inverse of cell stiffness in units of kPa$^{-1}$) and the dimensionless exponent $\beta$ representing cell fluidity with values ranging between 0 < $\beta$ < 1 pure elastic ($\beta = 0$) or viscous behavior ($\beta = 1$) and with the reference time $t_0$ was set to 1 s. The creep compliance $J(t)$ represents the ratio ($\gamma(t)/\sigma_0$) of the localized cellular strain $\gamma(t)$ induced by the applied stress from the magnetic tweezers $\sigma_0$, with $\gamma(t)$ taken as the radial bead displacement normalized over the bead radius $\gamma(t) = d(t)/r$ and the applied stress as $\sigma_0 = F_0/4\pi r^2$ taken as the applied force normalized over the bead cross sectional area. Compliance measurements for each BC cell line were collected from three independent experiments (MCF7 CTRL $n = 60$, MCF7 KRT80 $n = 34$, LTED CTRL $n = 41$, LTED KRT80 $n = 34$).

**Shearwave elastography.** All individuals involved were consented prior to measurements collection. All SWE was performed by a breast radiologist with more than 10-years' experience of performing Breast ultrasound and elastography on breast lesions. A state-of-the-art ultrasound scanner, Aplio i900 (Canon Medical Systems, Nasu, Japan) with the latest 2D SWE technology was used for this study. All SWE maps and calculations were obtained pre-biopsy. A good stand-off was used for superficial lesions and initially, continuous SWE mode ("multi-shot") was used to select the optimum plane and once this was stabilized, a higher energy SWE push-pulse ("one-shot" mode) was then utilized to obtain the final elastogram for calculations. Regions of interest (ROI) were placed within the center of the lesion, in the periphery and also within the adjacent normal breast tissue. This has been stored as raw data within the ultrasound systems which would enable any re-calculations as necessary.

**Reporting summary.** Further information on experimental design is available in the Nature Research Reporting Summary linked to this article.

## Data availability

RNA-seq expression profile can be found at: https://www.ncbi.nlm.nih.gov/geo/query/acc.cgi?acc=GSE125128. Single Cell RNA-seq data can be downloaded from ref. [38]. SREBP1 ChIP-seq are available upon request.

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

## Acknowledgements

We want to acknowledge and thank all patients and their families for the support and for donating the research samples. We thank Giacomo Corleone, Sung Pil Hong, Iros Barozzi, and Carlos Matellan for the technical assistance. The authors gratefully acknowledge infrastructure support from the Cancer Research UK Imperial Centre, the Imperial Experimental Cancer Medicine Centre and the National Institute for Health Research Imperial Biomedical Research Centre. L.M. was supported by a CRUK fellowship (C46704/A23110). Y.P. was supported by a CRUK Studentship (PS2099). A.J.F. and F.C. are funded by the Institute of Cancer Research, London (UK). F.C. is also funded by Worldwide Cancer Research (Grant 15-0273), Cancer Research UK (C57744/A22057) and the Ramon y Cajal Research Program (MINECO, RYC-2016-20352). We acknowledge Z. Magnani for his constructive comments on the manuscript.

## Author contributions

L.M. conceived the study. Y.P., A.J.F., A.R.M., A.U., A.C., P.M., M.F., C.D., J.H.C. performed all the experiments and analyses, A.L. performed Shearwave elastography, L.M., B.G., and F.C. performed analyses, S.S., G.P., and C.C. provided samples, A.U., L.M., and G.P. scored pathology sections, A.C. and A.R.H. performed stiffness measurements in cells, C.I. identified, collected and analyzed patient's samples. A.U., C.D., J.H.C., and N.P. performed tissue staining. L.M. and F.C. wrote the manuscript with contribution from all authors. All authors approved the manuscript.

## Additional information

**Competing interests:** The authors declare no competing interests.

