## [Peer Review File · Nature Communications]

Reviewers' comments:

Reviewer #1 (Remarks to the Author):

The study by Perone et al provides novel and potentially clinically relevant data linking endocrine resistance, epigenetic reprogramming including SREBP1 activation, and cytoskeletal changes resulting in increased breast cancer cell invasion. State-of-the-art approaches were used, and the study represents an elegant combination of mechanistic studies, and descriptive analysis of clinical samples. Overall, the study is well presented, and provides valuable insight for the greater breast cancer community.

Concerns that should be addressed are listed below:

- 1) The majority of studies are done in MCF-7 models. The study would be strengthened if the major observation and phenotypes could be studied in a few additional endocrine resistance models.
- 2) The authors make the point that the observed phenotypes are unique for AI treatment and not seen with Tam treatment. However, the difference between effects in Tam vs AI-resistant cell line models, and clinical samples seems minimal. For example, RNA seq data shows upregulation in both (Fig 1B), and there is a trend to upregulation in pre/post tam treated samples, too (Figure 1E). It might be good to de-emphasize treatment-specific effect, unless additional data can be provided that provide further supportive evidence.
- 3) Figure 1: Survival data for TCGA are very limited and frequently inaccurate, and thus Figure 1G should be replaced with METABRIC (and/or other larger data set) results.
- 4) The stiffness analysis is an important and novel aspect of the study, but should be expanded to solidify the data, and to connect it more closely to the overall topic of the manuscript. It would be important to add two analyses, which should be relatively easy to perform: 1) Study stiffness in KRT80 knock-down LTED cells, and 2) link the stiffness to AI resistance, for example through intra-tumoral analysis of stiffness in responders and non-responders.
- 5) Is there a correlation between cytokeratin 80 and SREBP1 in clinical samples, especially after endocrine therapy?
- 6) NGS data (e.g. single cell RNA seq; RNA seq with cytokeratin overexpression; ChIP-seq etc) should be made available, and links to data repository should be included into Methods section.

Minor:

- Line 74: The cell line models - untreated, non-invasive parental vs. invasive parental LTED - should be briefly explained in this current manuscript (in addition to providing the reference).

- The abstract states “Shear-wave elasticity imaging of prospective patients”. This should be changed to “tumors”, instead of “patients”.
- Check completeness of References (e.g. #3, 14, 16 and other are missing years and other information). Also, include Cheung et al PNAS 2016.
- Figure legend Fig 1F: Rephrase “is a bad prognostic marker”, as this is ambivalent.
- Suppl Figure 3: Details of data sources should be provided in figure legends
- It would be helpful for the readers if the manuscript would include a graphical presentation of type II-keratin TAD, with names of cytokeratins, mRNA FC LTED vs parental, and enhancers.
- The authors state that “we prospectively recruited 20 patients with suspected BC and performed 156 shear-wave elastography...”. Why is this “suspected” breast cancer? Was this analysis done before biopsy was taken?

Reviewer #2 (Remarks to the Author):

The manuscript by Perone et al. provides an interesting study on KRT80 up-regulation and invasive behaviour in cells that are resistant to aromatase inhibitors (AI). In resistant cells, SREBP1 binds at an enhancer in close proximity of the Krt80 gene, enhancer that appears to promote KRT80 expression. KRT80 overexpressing cells show an enhanced expression of cytoskeletal genes and an increased invasiveness, while its knockdown leads to decrease cell mobility. KRT80 expression is associated to poor prognosis in breast cancer patients. This study explores an attracting model where SREBP1 activation in AI resistant cells would promote cholesterol biosynthesis (described in previous work from the laboratory – Nguyen et al. 2015) and cell migration through KRT80 activation.

While a previous study of the Magnani group emphasised the link between AI resistance and epigenetics rearrangements (see above), this work propose to link an enhancer activation by SREBP1 with KRT80 activation, cell mobility and invasion. In general, the work presented is of significant relevance to the field, however the manuscript must be improved substantially regarding description and interpretation of the data. In particular the quality of the genomic analysis, the figures and their legends needs major improvement before publication.

Comments:

- The analysis of the genomic dataset is missing from the method section.
- Figure 1A and S4A refer at changes in H3K27ac at the TAD of KRT80. The snapshot presented in S4A should be presented and cited together with Figure 1A.
- Figure S4A, the scale of the ChIP-seq signal is missing.
- In the Figure S4A, H3K27ac appears at the background level in LTED cells. The conclusion that H3K27ac signal is increased in this cell line at the TAD of KRT80 is not in accordance with this figure.
- Could the authors explain the disparity between the two figures (1A, S4A) and how the datasets have been processed?
- ChIP-seq data analyses are not described in the text or the method section.
- Figure 1A (top), all cell type labels should appear on the chart of H3K27ac signal.
- Cell lines that are used in the study should be properly introduced in the text (eg. LTEDT, LTEDF, MCF7T and MCF7F).
- Accession numbers of the datasets presented must appear in the text and figure legends with appropriate citation when needed. (eg. the HiC dataset)
- Lines 83 to 85; 392 to 395: Single cell RNA-seq data suggests that KRT80 increase is driven by transcription. This conclusion could only be made when RNA-seq is compared to the absence of major cell division and apoptosis. These datasets should also be presented in supplementary figures.
- Figure 2B, the method used for motif analysis must be provided.
- Figure 2B, the authors should also provide the full analysis; what are the other motifs found? They should also add the statistical significance of the motif enrichments.
- Figure 2C, SREBP1 binding is clearly enhanced at HMGCR and E1-KRT80 enhancer, the author should add H3K27ac to these charts or in supplementary figures using the same genomic regions. If H3K27ac signal is not changing (MCF7 vs LTED), the authors should comment on that (see figure S4A).
- A comprehensive analysis of SREBP1 ChIP-seq has not yet been performed. How many peaks are in common between MCF7 and LTED cells (as at SREBP1 promoters) and how many differ (eg. HMGCR, E1-KMT80)? Is SREBP1 truly activated or actually re-distributed on the genome?
- Figure 2D, authors propose that SREBP1 control KRT80 at the transcription level; effects of shSREBP1 #1 and #2 should be also addressed at RNA levels.
- Figure 2F, the scale of mRNA expression of KRT80 is missing. No conclusion can be made with the actual presentation (changes appear quite marginal...)
- The Figure 3B can't be properly understood as presented. I suppose that the first 8 pictures are related to figure 3C first chart. Thus, the LTED legend would be missing in 3B.

- Figure 3B, could the authors comment on the changes observed in the colony between MC7- and MCF7+ KRT80-DKK at day 0?
- Figure 3C, the figure is sloppy and statistical bars are not properly aligned. Error bar information is missing in the legend.
- The legend and the descriptions of table S1 are missing.
- Figure 4, The authors propose an attracting model where SREBP1 would promote both invasion and survival:
 - o Up-regulation of many SREBP1 target genes is seen in AI resistant cells. Inhibition of SREBP1 translocation alters this, supporting that SREBP1 is regulating these genes (Nguyen et al. - 2015). However, it is not clear if SREBP1 is activated or if it is re-distributed in these cells. The authors should monitor SREBP1 translocation and phosphorylation status at least in MCF7 and LTED cells. This would allow concluding that SREBP1 is activated. Analysis of the SREBP1 ChIP-seq datasets would also help.
 - o While SREBP1 is required for KRT80 expression, the authors have not demonstrated that it is sufficient. The authors should discuss this or could improve their model by monitoring effects of over-expression of SREBP1 on KRT80 regulation.

Reviewer #3 (Remarks to the Author):

Summary:

The manuscript by Perone et al, aims to identify AI treatment resistance through the activation of transcriptional programs regulating cell stiffening and invasion. The authors complete a rigorous analysis of epigenetic changes in the TAD domain of Type II Keratins to identify KRT80 regulation in AI-resistant cells through the transcription factor SREBP1. This activation leads to enhanced stiffness at both the cellular and tumor level. The conclusion that increased stiffness is attributed to KRT80 localization with F-actin and focal adhesion presence is correlative and further experiments and quantification of the data are necessary to justify this conclusion. Overall, the manuscript has many novel findings, the impact of which advance the understanding of AI-resistance mechanisms. However, the large number of editorial mistakes in the manuscript and the need for additional experiments, quantification, and detailed description of methods prevents this paper from being publishable in its current form.

Major Revisions:

1) The manuscript would benefit from a careful and detailed review to ensure the figures and legends are properly labeled and referenced in the manuscript. Multiple errors in figure labeling and referencing in both the text and figure legends are incorrect, mislabeled and not described well enough. This made the manuscript very difficult to read and interpret in the current form. Several mistakes have been pointed out specifically in the bullet points below, however I strongly encourage additional review by authors or trusted colleague to ensure all mistakes are corrected.

2) Some of the conclusions were overstated, specifically in reference the role that KRT80 has on F-actin, lamellipodia formation and focal adhesion maturation. F-actin, KRT80 and focal adhesion intensity and the following quantification was not described. To conclude that these proteins are co-localized to the lamellipodia would require spatial quantification comparing cytoplasmic to lamellipodial regions of the cell. The quantification of F-actin and KRT80 may be specific to lamellipodial regions but the method of quantification is not described in enough detail to understand this point, details of the methodology need to be included. Along these lines, in Fig 1D actin is used as a loading control. If KRT80 expression leads to an increase in F-actin is it possible that total actin levels may also change. The authors should clarify whether there is a change in total actin levels compared to total protein or DNA before using actin as a loading control. It may be interesting to determine whether KRT80 increases total cellular actin or just enhances actin polymerization in the lamellipodia. Spatial quantification of F-actin in the lamellipodia and experiments to determine the ratio of G to F actin will help clarify this point. Similarly, quantification of focal adhesions is vague. Are you quantifying number or area of focal adhesions. Is this done for the whole field of view or only along the invasive front of the spheroid? This brings up another question, how do the authors rationalize the increase in focal adhesion formation with increased migration. Is there an increased rate of FA turnover with increased KRT80 expression?

3) KRT80 overexpressing cell spheroids do spread more than control cells, however it is unclear to this reviewer whether that is due to migration or a change in proliferation rates between the cell types. Cell proliferation should be determined for each of the cell lines for the duration of the experiment. Additionally, higher magnification images to highlight the initial spheroid size and subsequent migration or spheroid growth after 2 days in culture would help to convince the reviewer. Do single cells release from the spheroid cluster to invade or do they maintain cell-cell contacts? Images in Fig3 show some cells/cell clusters separated from the spheroid, are these the cells with the highest levels of KRT80? Furthermore, time lapse imaging of migration would greatly improve enhance the role of KRT80 in cell migration along with an alternative method to quantify cell migration, such as a 3D transwell assay or 2D scratch wound assay.

4) It would also be of interest to know if this phenotype is specific to KRT80 or whether another type II Keratin would have a similar effect. In Figure 1A the authors identify upregulation of NR4F1, KRT86 and KRT81 along with KRT80 in resistant/invasive cells. Would knockdown or overexpression of one of these other keratins result in a similar increase in stiffness, lamellipodia formation and migration?

Minor Revisions:

Figure 1:

- Fig 1A – the organization of the figure and description in the text/figure legend is confusing and difficult to follow. The Hi-C interaction map could be moved to supplemental allowing for a focus on the RNA-seq data and acetylation graphs.
- Fig 1D- later in the paper the authors describe that KRT80 influences actin formation, which could be due to either increased F-actin formation as well as increased actin expression. Thus, actin may be a poor choice for a loading control for this experiment.
- Is 1E the quantification of 1F? The description of 1E is so poor that I don't know what the data is showing, matched samples of pre vs post treatment or primary and metastatic sites?
- 1F figure legend refers to i, ii, and iii yet the images are not labelled in that way. We would also appreciate if zoom in images are shown with arrows showing areas where KRT80 becomes strongly cytoplasmic. Also, quantification of this observation would be powerful here. Please add error bars to all images.
- 1H should and 1I could be moved to supplemental.

Figure 2:

- In A, color is flipped with Seq lines
- Fig. 2 A, B, C and D be combined with Figure 1 to show KRT80 and SREBP1. This would allow for Figure 2 to focus on KRT80 effect on tissue stiffness.
- 2D why is there not a doublet for KRT80 as shown in figure 1?
- 2E image needs to be reoriented (or scale bar needs to be)
- 2E can you compare MCF7 control to LTED control? If there is not a difference, can you explain why that may be?
- 2F TCGA plot should be removed to supplemental as it distracts from the focus of the figure.
- 2G IHC quantification technique is unclear (% area of image?), a description of technique is desired.

Figure 3:

- 3B better labeling is necessary. Is the quantification in graph C correspond to the images in 3B? If so, the labeling on the images needs to be improved and match the graph in 3C. What are the numbers that are in the lower left? What is KRT80 DKK? This is unclear. Scale bars are not consistent.

- 3C error bar needs to be moved for first bar graphs.
- 3D enhanced brightness and contrast, add scale bar. Could you label the “invasive front” by having a line showing the boundary of the plug? Quantification of this would be desired summarizing this observation in multiple invasion assays. A high mag inset of the invasive front would also be helpful.
- 3E is poorly organized. Scale bars are not consistently placed. There isn’t a scale bar on the zoom in areas. The zoom in areas are not in the same orientation of the original image making it hard to follow. Arrows should be used to mark what you are describing. Better quantification of lamellipodia. Keep order of colors consistent.
- 3F should include an image of Focal adhesions to correspond with the quantification. There is no description of how this quantification is conducted, and must be added. It would be better to quantify focal adhesion number and average focal adhesion size. % area doesn’t tell us if there is more FA or just larger FA.

Figure 4:

- 4A should be placed in supplemental.
- 4B needs to have better labeling to highlight tumor suppressors genes and cytoskeletal associated genes.
- 4C please show other genes that are not UP in graph, this shows that all are upregulated but hard to compare to know if cytoskeletal genes are uniquely upregulated.
- 4D, there needs to be quantification of cortactin increase for IF, perhaps by western blot. Are these three images representing the same thing (3 images of MCF7 and 3 of MCF7-K80) or is there something different about the images?
- 4F Model should include only what was described in the paper for upper portion. Lower should be removed or referenced to provide reason for inclusion in the paper.

We want to thank all the reviewers for their constructive comments, we sincerely appreciate their time and commitment toward reviewing our manuscript. We have attempted to experimentally address all the queries and have reworked the text as necessary. The original comments are displayed in “normal and black font”. Our replies are reported in “italic and blue font”. We invite the reviewer to pay particular attention as figure numbers as the order has been changed to properly address all their concerns.

Reviewers' comments:

Reviewer #1 (Remarks to the Author):

The study by Perone et al provides novel and potentially clinically relevant data linking endocrine resistance, epigenetic reprogramming including SREBP1 activation, and cytoskeletal changes resulting in increased breast cancer cell invasion. State-of-the-art approaches were used, and the study represents an elegant combination of mechanistic studies, and descriptive analysis of clinical samples. Overall, the study is well presented, and provides valuable insight for the greater breast cancer community.

We are very happy that the reviewer finds our study novel and potentially relevant to patients. We have addressed her/his concern in details as follows:

1) The majority of studies are done in MCF-7 models. The study would be strengthened if the major observation and phenotypes could be studied in a few additional endocrine resistance models.

We appreciate the reviewer comment and agree that a larger panel of cells would provide even further strength to the manuscript. Where possible, we have used two models (MCF7 and T47D) to show KRT80 longitudinal upregulation in response to AI. We would like to point out that the extensive use of multidisciplinary tools, namely, epigenetic, single cell transcriptomics, 3D organoids and physical measurements has played a role in focusing on a single model for more details. We also want to point out that the prospective study in patients and the integration with all available public datasets shows that our results holds in general terms.

2) The authors make the point that the observed phenotypes are unique for AI treatment and not seen with Tam treatment. However, the difference between effects in Tam vs AI-resistant cell line models, and clinical samples seems minimal. For example, RNA seq data shows upregulation in both (Fig 1B), and there is a trend to upregulation in pre/post tam treated samples, too (Figure 1E). It might be good to de-emphasize treatment-specific effect, unless additional data can be provided that provide further supportive evidence.

We have revised the text according to the reviewer suggestions, and reduce emphasis on treatment-specific effects. For more details, see the new result paragraph (Page 3) and specifically lines 116-118.

3) Figure 1: Survival data for TCGA are very limited and frequently inaccurate, and thus Figure 1G should be replaced with METABRIC (and/or other larger data set) results.

We have revised the text and figures according to the reviewer suggestions. We now present METABRIC data with various subgroups and different follow ups (New figure 2C) and have relocated TCGA data in Supplementary figures (Supplementary Figure 1-2). We want to highlight that we have analyzed all the available large BC datasets with KRT80 expression (Metabric, TCGA and all GEO-depsited microarray re-normalized as a single large study, Supplementary Figure 1). We have also performed robust multivariate analysis to correct for clinical correlates and used all available endpoints (Supplementary Figure 1D)

4) The stiffness analysis is an important and novel aspect of the study, but should be expanded to solidify the data, and to connect it more closely to the overall topic of the manuscript. It would be important to add two analyses, which should be relatively easy to perform: 1) Study stiffness in KRT80 knock-down LTED cells, and 2) link the stiffness to AI resistance, for example through intra-tumoral analysis of stiffness in responders and non-responders.

We appreciate the reviewer comments and have tried, wherever possible, to satisfy her/his requests.

Point 1) We have now performed in vitro measurements as suggested by the reviewer. Data are presented in new Figure 4A (right panel). Silencing KRT80 in LTED cells induce a loss of stiffness in this model, as predicted by the reviewer.

Point 2) Unfortunately, performing additional stiffness measurements in actual patients, is extremely complex. The study proposed in 2) would involve an additional clinical study in which patients should be monitored for at least 5 years after stiffness measurements to determine time to relapse, as a proxy of drug resistance. Because Shearwave elastography is not routinely collected at diagnosis (nor at relapse), at least in our affiliated Hospital and more generally in the UK, it is impossible to use historical data to address the link between AI resistance and stiffness.

*Another potential study, as the reviewer suggested, would be perhaps to measure matched samples before and after neo-adjuvant AI therapy (as in the manuscript we suggest exposure to therapy **drives** SREBP1-KRT80 axis and leads to increase stiffness and possibly increased invasion). Stiffness measurements were performed through great collaborative work with the radiology department and involve a very laborious clinical set up. Neo-adjuvant treatment is not widely implemented in our hospital, thus limiting the number of samples we might be able to measure. Our data suggest that KRT80 upregulation is triggered by therapy but is not necessarily upstream of the drug-resistance phenotype (while being essential for the invasive phenotype). Necrosis after neo-adjuvant therapy, on the other hand, might confuse the results of a potential Shearwave neo-adjuvant study (necrosis might increase significantly the stiffness readout). Additionally, this study is complicated by the fact we would have to measure baseline stiffness and then measure it after treatment while correcting for changes in the tumor volume. Overall, these clinical studies would need a substantial level of funding and time.*

5) Is there a correlation between cytokeratin 80 and SREBP1 in clinical samples, especially after endocrine therapy?

*This is an important point and we thank the reviewer for raising it. The suggested analysis is complicated by the potential mechanism of action of SREBP1. We did not observe any difference in the expression level of SREBP1 in MCF7 and LTED (and in many other endocrine therapy resistant models), while KRT80 is dramatically altered. This apparent paradox stems from the actual SREBP1 activation cascade (see new model in New Figure 7F). The bulk of SREBP1 proteins is in the cytoplasm. Upon activation, the first cleaving event leads to SREBP1 accumulation in the Golgi. A second cleaving event releases SREBP1 to the nucleus. We are investigating these events in a separate project (from which the CHIP-seq results in new Figure 3 were extrapolated). The correct analysis in IHC and tissue would be to measure **intranuclear** SREBP1 levels in pre and post-AI treated samples and correlate that measurement to KRT80 levels. However, all commercially available antibodies are not optimized for this particular analysis. We are currently working toward developing antibodies specific for the intra-nuclear form of SREBP1. Only after these technical improvements it will become possible to test what the reviewer is suggesting. Indirect data supporting the reviewer's point can be found in Supplementary Figure 4D. Indeed, mRNA levels for SREBP1 canonical targets and KRT80 are strongly correlated in normal and breast cancer datasets, suggesting that SREBP1 activation in breast cancer can trigger KRT80 transcription.*

6) NGS data (e.g. single cell RNA seq; RNA seq with cytokeratin overexpression; CHIP-seq etc) should be made available, and links to data repository should be included into Methods section.

ChIP-seq for SREBP1 will be disclosed in a separate manuscript (MCF7, T47D, ZR75, LNCAP).

-Single cell RNA-seq data can be found here: "Raw sequencing data was deposited at the Gene Expression Omnibus (GEO) under accession number GSE119693. Reviewers can access the data using token mzgymeowzftneb and following the instructions provided at this link: <http://www.ncbi.nlm.nih.gov/geo/query/acc.cgi?acc=GSE122743>. These data were part of a separate manuscript (<https://www.biorxiv.org/content/early/2018/12/02/485136>, currently under review with Nature Communications). The link has been also cited in the text.

-RNA-seq after KRT80 over-expression can be found here: <https://www.ncbi.nlm.nih.gov/geo/query/acc.cgi?acc=GSE125128>. Data sharing statement has been included in the manuscript.

Minor:

- Line 74: The cell line models - untreated, non-invasive parental vs. invasive parental LTED - should be briefly explained in this current manuscript (in addition to providing the reference).

Text was modified based on reviewer's suggestion. The revised section can be found at line 78-81.

- The abstract states “Shear-wave elasticity imaging of prospective patients”. This should be changed to “tumors”, instead of “patients”.

Text was modified based on reviewer’s suggestion. The revised section can be found at line 211-213.

- Check completeness of References (e.g. #3, 14, 16 and other are missing years and other information). Also, include Cheung et al PNAS 2016.

References were checked and fixed where needed. See line 379.

- Figure legend Fig 1F: Rephrase “is a bad prognostic marker”, as this is ambivalent. *Text was modified based on reviewer’s suggestion, see line 575*

- Suppl Figure 3: Details of data sources should be provided in figure legends

Text was updated based on reviewer’s suggestion. Figure has been updated to Supplementary Figure 1-2.

- It would be helpful for the readers if the manuscript would include a graphical presentation of type II-keratin TAD, with names of cytokeratins, mRNA FC LTED vs parental, and enhancers.

Bird’s eye view of the Keratin TAD was offered in Supplementary Figure 4, but now moved in new Figure 1B. RNA-seq counts is in Figure 1A and shown as an heatmap. A more detailed map on enhancers in MCF7 and LTED is presented in figure 3. Overall, we have updated Figure 1 and 3 to convey the sense of 2D topology of the keratin genes distribution. Supplementary figure 3 and 4 also shows ChIA-PET derived plots to show interactions between enhancer and promoters.

- The authors state that “we prospectively recruited 20 patients with suspected BC and performed 156 shear-wave elastography...”. Why is this “suspected” breast cancer? Was this analysis done before biopsy was taken?

Text was updated based on reviewer’s comments. Biopsies were taken after elastography (hence, suspected, as the patients had only received an ultrasound prior to elastography). Nevertheless, suspected cancers were all confirmed by histopathology before being included in the study.

Reviewer #2 (Remarks to the Author):

The manuscript by Perone et al. provides an interesting study on KRT80 up-regulation and invasive behaviour in cells that are resistant to aromatase inhibitors (AI). In resistant cells, SREBP1 binds at an enhancer in close proximity of the Krt80 gene, enhancer that appears to promote KRT80 expression. KRT80 overexpressing cells show an enhanced expression of cytoskeletal genes and an increased invasiveness, while its knockdown leads to decrease cell mobility. KRT80 expression is associated to poor prognosis in breast cancer patients. This study explores an attracting model where SREBP1 activation in AI resistant cells would promote cholesterol biosynthesis (described in previous work from the laboratory – Nguyen et al. 2015) and cell migration through KRT80 activation.

While a previous study of the Magnani group emphasised the link between AI resistance and epigenetics rearrangements (see above), this work propose to link an enhancer activation by SREBP1 with KRT80 activation, cell mobility and invasion. In general, the work presented is of significant relevance to the field, however the manuscript must be improved substantially regarding description and interpretation of the data. In particular the quality of the genomic analysis, the figures and their legends needs major improvement before publication.

We thank the reviewer for finding the study interesting and relevant to the field. We agree that there were several inconsistencies in the first submission and we hope to have rectified all these issues in the present version of the manuscript.

Comments:

- The analysis of the genomic dataset is missing from the method section.

We apologize for the missing information, we assume the review means the analysis of the TAD locus and ChIP-seq. Additional information has been added in the method section under the TAD analysis section and under the Library preparation and ChIP-seq data analysis. We have also added information on the single-cell RNA-seq (and reference the source of the data). See lines 742-752 and 866-873

- Figure 1A and S4A refer at changes in H3K27ac at the TAD of KRT80. The snapshot presented in S4A should be presented and cited together with Figure 1A.

Text and figures were updated based on reviewer's comments. Please see the new Figure 1.

- Figure S4A, the scale of the ChIP-seq signal is missing.

Figure was updated based on reviewer's comments

- In the Figure S4A, H3K27ac appears at the background level in LTED cells. The conclusion that H3K27ac signal is increased in this cell line at the TAD of KRT80 is not in accordance with this figure.

We apologize for the misunderstanding. The signal change at TAD level has been calculated through ranking all TADs averaged signals genome-wide, to account for different efficiencies in ChIP-seq. Basically the TAD locus in LTED is amongst the TAD with the strongest change in bulk acetylation when comparing the two cell lines. The signal appears low as we had to accommodate the scale to cell lines with even stronger signal at this particular locus (LTEDT and F plus MCF7 Fulvestrant). To increase the confidence of this observation we have performed a series of focused ChIP analysis on the KRT80 enhancer using H3K27ac plus two additional marks (H3K4me1 and me2, new Figure 1D). These data support an increased activation of KRT80 enhancer in LTED cells (see lines 81-86).

- Could the authors explain the disparity between the two figures (1A, S4A) and how the datasets have been processed?

We respectfully refer the reviewer to the comment above and additional method. We believe there is no discrepancy in the data. Briefly, we normalize ChIP-seq data by ranking the signals within each cell line. This allow comparisons of ranks rather than total signal (that would also be driven by different efficiencies between experiments). The KRT TADs in LTED accumulates much more signal (when accounting for overall signal) compared to the same locus in MCF7 (where globally, the ChIP protocol resulted in higher overall signal).

- ChIP-seq data analyses are not described in the text or the method section.

Text and figures were updated based on reviewer's comments. We have now added an additional section to described SREBP1 ChIP-SEQ data. H3K27ac ChIP-seq were obtained from our previous publication and cited accordingly. See lines 788-819 and 821-837. The ranking analysis was described previously (Nguyen, Nat. Comm 2015) and cited accordingly in the text (see line 76).

- Figure 1A (top), all cell type labels should appear on the chart of H3K27ac signal.

Figure was updated based on reviewer's comments. See new Figure 1A.

- Cell lines that are used in the study should be properly introduced in the text (eg. LTEDT, LTEDF, MCF7T and MCF7F).

Please, see response to reviewer 1. Cells have been now briefly introduced in the text (see lines 78-81).

- Accession numbers of the datasets presented must appear in the text and figure legends with appropriate citation when needed. (eg. the HiC dataset)

Text and figures were updated based on reviewer's comments. External data have been referenced, with their accession numbers, in figure legends and/or material and methods. References, when appropriate, have been added.

- Lines 83 to 85; 392 to 395: Single cell RNA-seq data suggests that KRT80 increase

is driven by transcription. This conclusion could only be made when RNA-seq is compared to the absence of major cell division and apoptosis. These datasets should also be presented in supplementary figures.

We appreciate the reviewer comments and performed additional experiment to support our statement. We then performed a continuous imaging proliferation assay so we could monitor individual cells during the course of estrogen acute deprivation (new Figure 1D). These data show that there is major significant change in cell number, but more importantly, there is no significant cell death, suggesting that over 90% of the cells measured in the single-cell RNAseq experiment should be conserved before and after treatment thus reinforcing the notion that KRT80 is transcriptionally upregulated at the single cell level rather than increased through a selection process.

- Figure 2B, the method used for motif analysis must be provided.

Text and figures were updated based on reviewer's comments. An additional section in M&M has been added. This figure is now labelled Figure 3D.

- Figure 2B, the authors should also provide the full analysis; what are the other motifs found? They should also add the statistical significance of the motif enrichments.

Full analysis was published previously (Nguyen, Nat. Comm. 2015). More information has been added to the text (see lines 177). We would suggest that the addition of the other footprints might not be useful for the current manuscript, as each analysis reports several potential TF binding sites for each individual footprint. Nonetheless, we have mentioned that more than one footprint was found in the analysis (line 179). The actual ChIP-seq results prove without doubt that SREBP1 binding at this enhancer occurs uniquely in LTED models (MCF7 and T47D).

- Figure 2C, SREBP1 binding is clearly enhanced at HMGCR and E1-KRT80 enhancer, the author should add H3K27ac to these charts or in supplementary figures using the same genomic regions. If H3K27ac signal is not changing (MCF7 vs LTED), the authors should comment on that (see figure S4A).

We would kindly refer the reviewer to the new Figure 1C. As currently hinted by the reviewer, H3K27ac changes between MCF7 and LTED, while a similar change does not occur for other histone marks.

- A comprehensive analysis of SREBP1 ChIP-seq has not yet been performed. How many peaks are in common between MCF7 and LTED cells (as at SREBP1 promoters) and how many differ (eg. HMGCR, E1-KMT80)? Is SREBP1 truly activated or actually re-distributed on the genome?

We are currently studying the mechanism of activation of SREBP1 in the context of endocrine therapy. The general analyses will be published in due course (aiming at end of the year), but we can anticipate to this reviewer that SREBP1 is genuinely

activated in LTED models, not redistributed. Indeed, the only two peaks called in MCF7 are the one shown in Figure 3C at SREBP1 promoters (see Venn diagrams attached to this reviewer last comments). The mechanisms behind SREBP1 activation are complex in nature and will require an ad hoc publication. If the reviewer wishes so, we can upload the data as an embargo dataset and share them with a password-protected access. Clearly, we are invested in understanding how SREBP1 is regulated and are actively investigating it. However, this avenue represent an independent research project and we decided to write it up in a separate manuscript.

- Figure 2D, authors propose that SREBP1 control KRT80 at the transcription level; effects of shSREBP1 #1 and #2 should be also addressed at RNA levels.

Text and figures were updated based on reviewer's comments. New Figure 3F shows changes in transcriptional level for KRT80 after shSREBP1.

- Figure 2F, the scale of mRNA expression of KRT80 is missing. No conclusion can be made with the actual presentation (changes appear quite marginal...)

Text and figures were updated based on reviewer's comments. An improved analysis was performed using TCGA and Affymetric RNA-based analyses. New graph and statistics associated with it are shown in new Supplementary Figure 6

- The Figure 3B can't be properly understood as presented. I suppose that the first 8 pictures are related to figure 3C first chart. Thus, the LTED legend would be missing in 3B.

We apologize for the mislabelling and the confusion. The reviewer is right, we have properly reworked this figure to increase its legibility (see new Figure 5A-D). Text was amended in lines 235-240

- Figure 3B, could the authors comment on the changes observed in the colony between MC7- and MCF7+ KRT80-DKK at day 0?

This is an interesting observation; however, I fear the image is misleading as after considering several different organoids we could not appreciate consistent qualitative differences in organoids formation We have updated the figure now to reflect this.

- Figure 3C, the figure is sloppy and statistical bars are not properly aligned. Error bar information is missing in the legend.

We apologize for the mislabelling. Text and figures were update based on reviewer's comments. Please see new Figure 3C and its associated figure legend.

- The legend and the descriptions of table S1 are missing.

Text and figures were update based on reviewer's comments.

- Figure 4, The authors propose an attracting model where SREBP1 would promote both invasion and survival:

o Up-regulation of many SREBP1 target genes is seen in AI resistant cells. Inhibition of SREBP1 translocation alters this, supporting that SREBP1 is regulating these genes (Nguyen et al. - 2015). However, it is not clear if SREBP1 is activated or if it is re-distributed in these cells. The authors should monitor SREBP1 translocation and phosphorylation status at least in MCF7 and LTED cells. This would allow concluding that SREBP1 is activated. Analysis of the SREBP1 ChIP-seq datasets would also help.

We fully agree with the referee and we are working on this specific subject (see also previous comment). It is very clear from experiment done in MCF7, T47D, ZR75 and LNCAP that SREBP1 is activated after hormone deprivation. Please see the confidential figures attached below

Figure for reviewer: SREBP1 ChIP-seq (three replicates consensus) in MCF7 and AI resistant LTEDs

Figure for reviewer: SREBP1 ChIP-seq (three replicates consensus) in MCF7 with short term hormone depletion or stimulation

However, we think this work should be published separately as we are investigating how SREBP1 senses estrogen/androgens, how long does it take to have full activation, and in which subpopulations this happens. These three lines of research require an independent manuscript and we believe they should be kept distinct from the current story in which we prove that SREBP1 controls KRT80 through direct activation of its enhancer. We have updated the model to improve the interpretation of the manuscript.

o While SREBP1 is required for KRT80 expression, the authors have not demonstrated that it is sufficient. The authors should discuss this or could improve their model by monitoring effects of over-expression of SREBP1 on KRT80 regulation.

This is a critical point but will need to be addressed in a separate manuscript as discussed in the previous comment. However, we agree that we should have been more explicit in stating that SREBP1 is required but not sufficient. We have amended the discussion to reflect this, see lines 323-324.

Reviewer #3 (Remarks to the Author):

Reviewer 3:

Summary:

The manuscript by Perone et al, aims to identify AI treatment resistance through the activation of transcriptional programs regulating cell stiffening and invasion. The authors complete a rigorous analysis of epigenetic changes in the TAD domain of Type II Keratins to identify KRT80 regulation in AI-resistant cells through the transcription factor SREBP1. This activation leads to enhanced stiffness at both the cellular and tumor level. The conclusion that increased stiffness is attributed to KRT80 localization with F-actin and focal adhesion presence is correlative and further experiments and quantification of the data are necessary to justify this conclusion. Overall, the manuscript has many novel findings, the impact of which advance the understanding of AI-resistance mechanisms. However, the large number of editorial mistakes in the manuscript and the need for additional experiments, quantification, and detailed description of methods prevents this paper from being publishable in its current form.

We are very happy that the reviewer finds our study very relevant and that it will impact in the current understanding of AI-resistance mechanisms. We acknowledge that there were several points for improvement, in particular organization of figures, editorial mistakes and lack of thorough information on methodology. In this revised manuscript, we have addressed these major issues as well as provided additional experiments and information on cytoskeletal rearrangements driven by KRT80 in the system.

Major Revisions:

1) The manuscript would benefit from a careful and detailed review to ensure the figures and legends are properly labeled and referenced in the manuscript. Multiple errors in figure labeling and referencing in both the text and figure legends are incorrect, mislabeled and not described well enough. This made the manuscript very difficult to read and interpret in the current form. Several mistakes have been pointed out specifically in the bullet points below, however I strongly encourage additional review by authors or trusted colleague to ensure all mistakes are corrected.

We apologize for the editorial defects that the reviewer highlights. In this updated manuscript we have thoroughly revised the text, legends, figures and methodology and amended these issues. We are confident that the manuscript is now clearer and better organized, and results can be better interpreted. All the specific points have been addressed; further information can be found below.

2) Some of the conclusions were overstated, specifically in reference the role that KRT80 has on F-actin, lamellipodia formation and focal adhesion maturation. F-actin, KRT80 and focal adhesion intensity and the following quantification was not described. To conclude that these proteins are co-localized to the lamellipodia would require spatial quantification comparing cytoplasmic to lamellipodial regions of the cell. The quantification of F-actin and KRT80 may be specific to lamellipodial regions but the method of quantification is not described in enough detail to understand this point, details of the methodology need to be included.

We now provide further analysis and description in the text and methods section to support the role of KRT80 in actin organization. When discussed in the text, we describe that “KRT80 network was prominent at the leading edge of cells, usually localized at or annexed to actin-rich lamellipodium-like structures” (line 254). This statement is now backed by additional spatial analysis. Following the reviewer’s suggestion, we have now performed spatial quantifications of F-actin intensity at the cell cortex, lamellipodia and cytosol (Figure 6B-D). This analysis indicates that, whereas global F-actin levels do not vary in KRT80^{low} vs KRT80^{high} cells, there is a clear effect on lamellipodial F-actin levels upon KRT80 expression (Figure 6C&D). In addition, we have performed morphological analyses in leading cells (i.e. cells at the border of clusters) and observed that KRT80 expression has a direct impact in lamellipodia formation.

Along these lines, in Fig 1D actin is used as a loading control. If KRT80 expression leads to an increase in F-actin it is possible that total actin levels may also change. The authors should clarify whether there is a change in total actin levels compared to total protein or DNA before using actin as a loading control.

As explained before, we do not observe changes in global F-actin levels upon KRT80 expression but rather a reorganization of F-actin structures. Thus, KRT80 expression is associated with an increase of lamellipodial structures and a slight decrease of F-actin at the cortex and the rest of the cytosol.

It may be interesting to determine whether KRT80 increases total cellular actin or just enhances actin polymerization in the lamellipodia. Spatial quantification of F-actin in the lamellipodia and experiments to determine the ratio of G to F actin will help clarify this point.

As previously explained, we now provide quantification of F-actin in different regions of the cell and observe an increase in F-actin associated to lamellipodia, but no changes in total F-actin (New Figure 6C&D). In addition, we have determined the F/G ratio in MCF7 and LTED cells and observe no major differences (Rebuttal Figure 1 for the reviewer).

Rebuttal Figure 1: Western blots showing F-actin (top) and G-actin (bottom) levels in MCF7 and LTED cells. Graph shows the quantification of the relative intensity of F-actin vs G-actin in MCF7 and LTED cells. F- and G-actin were separated by differential sedimentation following the approach described in Calvo et al, Cell Reports 2015 (doi: 10.1016/j.celrep.2015.11.052).

Similarly, quantification of focal adhesions is vague. Are you quantifying number or area of focal adhesions. Is this done for the whole field of view or only along the invasive front of the spheroid?

Analysis of focal adhesions is now described in more detail in the methods section. Regarding the specific questions of the reviewer, in the previous version of the manuscript we provided quantification of focal adhesion area along the invasive front of the cluster. We apologize if this was not sufficiently clear in the text/legend/methods; we have now provided further details for this. In addition, we now also provide quantification of number of focal adhesions per individual cell (also, only cells at the margin of clusters, new Figure 5F).

This brings up another question, how do the authors rationalize the increase in focal adhesion formation with increased migration. Is there an increased rate of FA turnover with increased KRT80 expression?

This is a very interesting question. Focal adhesions (FAs) are the points of contact of a cell with the substrate, and therefore more or bigger FAs is intuitively associated with a stronger adhesion that may difficult migration. However, FAs are critical for the generation of the tensional forces required for cytoskeletal rearrangements that elicit the cell shape changes and generation of structures required for migration. In particular, the forward movement of the lamellipodium requires focal adhesion growth and maturation at the leading edge to propel the cell forward (“Cell adhesion: integrating cytoskeletal dynamics and cellular tension”, Parsons et al Nature Reviews Molecular Cell Biology 2010, doi: 10.1038/nrm2957). On the other hand, focal adhesion disassembly occurs mainly at the rear where the cell needs to detach to move the cell body forward. We rationalize that the increased focal adhesion at the leading-edge results from the reorganization of F-actin and is a direct consequence of the generation of lamellipodia-like structures. Thus, the novel network of KRT80 positive filaments is promoting cell motility and invasive behaviors by reinforcing the organization of the lamellar actin fiber network and the stability of FAs.

3) KRT80 overexpressing cell spheroids do spread more than control cells, however it is unclear to this reviewer whether that is due to migration or a change in proliferation rates between the cell types. Cell proliferation should be determined for each of the cell lines for the duration of the experiment.

This is a sensible question and we have addressed it in a couple of ways. First of all, the organoids are embedded in Matrigel on all side and we have previously recorded invasion using continuous imaging. Two movies are provided to the reviewers from the original Nguyen 2015 Nat. Comm. publication that support how cell movement rather than cell proliferation, underlie organoids expansion.

MCF7

LTED

In addition, we have performed a series of experiments using CMFDA live labelling. At the concentration used in these experiments, fluorescence is lost in just 2-3 cell division. The images in figure 4E-G show how invading cells retain the dyes, further demonstrating that these cells actively moved rather than divide at the border.

Additionally, higher magnification images to highlight the initial spheroid size and subsequent migration or spheroid growth after 2days in culture would help to convince the reviewer.

These data are now provided in Figure 4E-G in addition to the movie themselves.

Do single cells release from the spheroid cluster to invade or do they maintain cell-cell contacts? Images in Fig3 show some cells/cell clusters separated from the spheroid, are these the cells with the highest levels of KRT80?

Cells remain in contact while invading (please see additional data and movies). Nonetheless, upon KRT80 removal, cell to cell contact appears to become more efficient, and some cells appear to bubble out of the organoid (more evident in the siRNA experiments, Supplementary Figure 9F). These cells fail to proliferate or invade. These data are in good agreement with KRT80 potential role in cell to cell adhesion (Langbein 2010) as KRT80 is also found in desmosomes (see attached picture from Protein Atlas obtained in A-431 epithelial cells (KRT80 in green, same antibody used in our study, Microtubules in red and DAPI in blue)

Furthermore, time lapse imaging of migration would greatly improve enhance the role of KRT80 in cell migration along with an alternative method to quantify cell migration, such as a 3D transwell assay or 2D scratch wound assay.

Original movies were taken using a Metamorph system, currently unavailable to our lab. We have attempted to take live imaging of organoids using Incucyte technology (same kit used to do live-imaging in other experiments). However, this strategy is impaired by the single-automatic focus set up of the kit and the impossibility to select areas prior to the experiment. Nonetheless, with the new images and movies we show that: LTED invasion is directed by active motions and we can completely exclude proliferation at the border. Ablation of KRT80 blocks invasion and KRT80 over-expression increases invasion. Overall, we hope these data will convince the reviewer. These data are now provided in Figure 5E-G in addition to the movie themselves.

Additionally, we have tested siKRT80 (less efficient than stable shRNA) on 2D migration assay using a real-time monitoring kit (iCelligence system). Keratin80 downregulation was effective in significantly slowing down migration on 2D substrates (scratch assay, Supplementary Figure 8E)

4) It would also be of interest to know if this phenotype is specific to KRT80 or whether another type II Keratin would have a similar effect. In Figure 1A the authors identify upregulation of NR4F1, KRT86 and KRT81 along with KRT80 in resistant/invasive cells. Would knockdown or overexpression of one of these other keratins result in a similar increase in stiffness, lamellipodia formation and migration?

Although interesting, we believe this point would not add to the manuscript. NR4A1 is a nuclear receptor and not a Keratin, moreover it is upregulated only in MCF7 Fulvestrant resistant and not in the invasive AI lines. KRT86 and 81 are the only other Keratins upregulated in AI resistant lines, but they have no prognostic significance in breast cancer (new Figure 2E). KRT80 on the other hand is emerging as a prognostic marker in many other cancers (see reviewer figure below) suggesting it might have a more conserved role in driving invasive phenotypes.

COAD Colon Cancer

Low Grade Glioma

Minor Revisions:

Figure 1:

- Fig 1A – the organization of the figure and description in the text/figure legend is confusing and difficult to follow. The Hi-C interaction map could be moved to supplemental allowing for a focus on the RNA-seq data and acetylation graphs.

We have edited Figure 1 according to feedback from all reviewers. We hope the new version is much clearer. In addition, we have reworked the figure legend as well.

- Fig 1D- later in the paper the authors describe that KRT80 influences actin formation, which could be due to either increased F-actin formation as well as increased actin expression. Thus, actin may be a poor choice for a loading control for this experiment.

As previously explained, we now provide quantification of F-actin in different regions of the cell and observe an increase in F-actin associated to lamellipodia, but no changes in total F-actin (New Figure 6C&D).

- Is 1E the quantification of 1F? The description of 1E is so poor that I don't know what the data is showing, matched samples of pre vs post treatment or primary and metastatic sites?

Figures have now been edited and more information has been added (Now in Figure 2). Figure Legend has also been edited. Old figure 1E was the comparison of KRT80 positivity in three different longitudinal datasets of breast cancer. Old Figure 1F is KRT80 staining in several breast related pathologies.

- 1F figure legend refers to i, ii, and iii yet the images are not labelled in that way. We would also appreciate if zoom in images are shown with arrows showing areas where KRT80 becomes strongly cytoplasmic. Also, quantification of this observation would be powerful here. Please add error bars to all images.

We apologize for the confusing set up. We have reworked the figure and added additional pictures from the supplementary data to add clarity. We have also added arrows as requested by the reviewers. We appreciate that quantification would be a

bonus, but these qualitative pictures were taken and evaluated by trained pathologists and were not collected for quantitative analysis (as IHC is not particularly suitable for this approach). We have now added bars to all figures.

- 1H should and 1I could be moved to supplemental.

Figure have now been moved to Figure 3 to complete the epigenetic section of the manuscript.

Figure 2:

- *In A, color is flipped with Seq lines*
We apologize for the mistake, it has now been rectified.
- *Fig. 2 A, B, C and D be combined with Figure 1 to show KRT80 and SREBP1. This would allow for Figure 2 to focus on KRT80 effect on tissue stiffness.*

Many thanks for the suggestion, we have completely revised the figure order (from 4 to 7) to accommodate new data and hopefully increase the clarity. We hope the new layout is agreeable to this reviewer.

- 2D why is there not a doublet for KRT80 as shown in figure 1?

Many thanks for this comment. We should have specified in more details that the western blot in new figure 1G were obtained using a proprietary antibody (Langbein et al. 2010). In this particular manuscript they have validated the antibody and describe in details the aspecific band (doublet). Western blot in new Figure 3G have been obtained using the Protein Atlas antibody. This is the reason behind the disappearance of the doublet. Apologies again for the confusion.

- 2E image needs to be reoriented (or scale bar needs to be)

Figure was updated. Please see new Figure 4A

- 2E can you compare MCF7 control to LTED control? If there is not a difference, can you explain why that may be?

The reviewer is absolutely correct that it does not appear to be a significant difference between MCF7 and LTED control, and this is somewhat confusing. These experiments are carried over in batch plates/sensors and therefore it is not possible to draw conclusions by comparing different experiments. While we could control for matched cells with or without KRT80, the plate could not accommodate 4 different lines thus we focused on actual KRT80 manipulation within the same genotype/cell type. Nonetheless, the addition of shRT80 experiments now further demonstrate that KRT80 manipulation directly control cell stiffness.

- 2F TCGA plot should be removed to supplemental as it distracts from the focus of the figure.

We have now removed the panel (New Figure 4B) and updated the figure (New Supplementary Figure 6)

- 2G IHC quantification technique is unclear (% area of image?), a description of technique is desired.

We have updated the figure and the figure legend to answer to this query. New Figure 4D now shows that % is referred to the number of cells stained positive within each section considered for the analysis.

Figure 3:

- 3B better labeling is necessary. Is the quantification in graph C correspond to the images in 3B? If so, the labeling on the images needs to be improved and match the graph in 3C. What are the numbers that are in the lower left? What is KRT80 DKK? This is unclear. Scale bars are not consistent.

We apologize for the confusion and we have updated the figure according to reviewers' comments. Figures with quantification are more explicit now and have been re-arranged in new Figure 5 A-D. Labelling has been updated. Number in the lower left panel were removed as they are redundant with the actual quantification. KRT80-DKK is a tagged version of KRT80 and DKK was used to cross validate KRT80 over-expression (Supplementary Figure 8B). Bars scale have been updated.

- 3C error bar needs to be moved for first bar graphs.

Figure has been updated according to reviewers' comments. See Figure 5D

- 3D enhanced brightness and contrast, add scale bar. Could you label the "invasive front" by having a line showing the boundary of the plug? Quantification of this would be desired summarizing this observation in multiple invasion assays. A high mag inset of the invasive front would also be helpful.

Figure has been updated according to reviewers' comments. Higher magnification pictures can be found in Supplementary Figure 10. Invasion front analysis is now shown in new Figure 5H in the new CMFDA-organoids analysis.

- 3E is poorly organized. Scale bars are not consistently placed. There isn't a scale bar on the zoom in areas. The zoom in areas are not in the same orientation of the original image making it hard to follow. Arrows should be used to mark what you are describing. Better quantification of lamellipodia. Keep order of colors consistent.

We sincerely apologize for the poor editing and organization of this (and other figures). We have followed the reviewer's suggestions and made extensive changes to improve the organization of this figure, which we hope is now clearer. In particular, we have divided the original figure in two, so zoom up images are now separated (Figure 6A&B). This allowed for a better and more consistent organization of the figure. As suggested by the reviewer, scale bars are now consistently placed in original and zoom up areas. In all cases, orientation has been kept, and colors and

scales are consistent. In addition, we have included symbols to mark specific structures discussed in the text, and added a line scan analysis to illustrate the differences in KRT80 and F-actin staining across the cell cortex, lamellipodia and cytosol. Furthermore, we have performed additional spatial analyses of F-actin staining in these different regions (Figure 6C&D). In all cases, clear description of image analysis quantification is provided in the methods section.

- 3F should include an image of Focal adhesions to correspond with the quantification. There is no description of how this quantification is conducted, and must be added. It would be better to quantify focal adhesion number and average focal adhesion size. % area doesn't tell us if there is more FA or just larger FA.

We thank the reviewer for these suggestions. We have now included images of focal adhesion (i.e. pY118-paxillin) staining in the main Figures (Figure 6F). Just to clarification, previous analyses were of individual focal adhesion size (as suggested by the reviewer) and not FA % area; we apologize for not making it clearer before. In addition, we have included a quantification of focal adhesion number as suggested by the reviewer. We now provide clear details on how these quantifications were performed in the methods section.

Figure 4:

- 4A should be placed in supplemental.

We believe this figure should remain to highlight the experimental design and the robustness of the analysis.

- 4B needs to have better labeling to highlight tumor suppressors genes and cytoskeletal associated genes.

Figure has been updated according to reviewers' comments. See new Figure 7B

- 4C please show other genes that are not UP in graph, this shows that all are upregulated but hard to compare to know if cytoskeletal genes are uniquely upregulated.

Owing to space limitation, we cannot label ALL genes in the Figure. However, we have added a Supplementary Table 2 with the Upregulated Gene List and additional statistical analysis to support our finding that genes up-regulated upon K80 overexpression are mainly related to cytoskeletal processes.

- 4D, there needs to be quantification of cortactin increase for IF, perhaps by western blot. Are these three images representing the same thing (3 images of MCF7 and 3 of MCF7-K80) or is there something different about the images?

Again, we apologize for the poor clarity and organization of this figure. Indeed, the original figures corresponded to the same thing (i.e. 3 images of MCF7 and 3 of MCF7-K80); the idea was to illustrate that the observation was a general feature. For the sake of clarity, we now just show 1 image per condition. In addition, we now

provide quantification of cortactin fluorescence intensity (New Figure 7D). We also provide details on this quantification in the methods section.

- 4F Model should include only what was described in the paper for upper portion. Lower should be removed or referenced to provide reason for inclusion in the paper.

Figure has been updated according to reviewers' comments. A reference to the original manuscript has been added.

REVIEWERS' COMMENTS:

Reviewer #1 (Remarks to the Author):

My concerns have been appropriately addressed.

Reviewer #2 (Remarks to the Author):

As previously mentioned, Perone et al. provides a comprehensive and rigorous analysis of epigenetic changes at the KRT80 locus and SREBP1-dependent activation of KRT80 in AI resistant cells. The authors further show that KRT80 is involved in cell migration control.

The revised version of the manuscript accommodates all criticisms I raised in my review. I recommend the manuscript to be published now in Nature communications.

Minor point:

- A supplementary figure showing the H3K27ac ChIP-seq tracks of MCF7 and LTED cells (shown in Figure 1B) at an appropriate scale would strengthen the observation that the signal of this histone mark is enhanced in LTED. This is convincingly confirmed in figure 1C.

Reviewer #3 (Remarks to the Author):

The authors have satisfactorily addressed the suggestions of the previous reviews. The additional quantification, clear description of methods and results and overall organization of the manuscript have much improved. I recommend the manuscript for publication.

We would like to thank the editor and all the reviewers once again for their constructive comments, we sincerely appreciate their time and commitment toward reviewing our manuscript. All the queries have been addressed point-by-point and have reworked the text as necessary. The original comments are displayed in “normal and black font”. Our replies are reported in “*italic and blue font*”.

REVIEWERS REQUESTS:

Reviewer #1 (Remarks to the Author):

My concerns have been appropriately addressed.

Reviewer #2 (Remarks to the Author):

As previously mentioned, Perone et al. provides a comprehensive and rigorous analysis of epigenetic changes at the KRT80 locus and SREBP1-dependent activation of KRT80 in AI resistant cells. The authors further show that KRT80 is involved in cell migration control.

The revised version of the manuscript accommodates all criticisms I raised in my review. I recommend the manuscript to be published now in Nature communications.

Minor point:

- A supplementary figure showing the H3K27ac ChIP-seq tracks of MCF7 and LTED cells (shown in Figure 1B) at an appropriate scale would strengthen the observation that the signal of this histone mark is enhanced in LTED. This is convincingly confirmed in figure 1C.

We appreciate that the reviewer would want to see an appropriately scaled version of Figure 1B. However, we think that showing re-sized scales would be misleading to the general audience (as we would consider this cherry picking). As the reviewer also suggest that the figure 1C convincingly shows difference, we would rather avoid inflating the results by showing cherrypicked cell lines.

Reviewer #3 (Remarks to the Author):

The authors have satisfactorily addressed the suggestions of the previous reviews. The additional quantification, clear discription of methods and results and overall organization of the manuscript have much improved. I recommend the manuscript for publication.